# On the Behaviour of an AC Induction Motor as Sensor for Condition Monitoring of Driven Rotary Machines

**DOI:** 10.3390/s23010488

**Published:** 2023-01-02

**Authors:** Mihaita Horodinca, Neculai-Eduard Bumbu, Dragos-Florin Chitariu, Adriana Munteanu, Catalin-Gabriel Dumitras, Florin Negoescu, Constantin-Gheorghe Mihai

**Affiliations:** Faculty of Machines Manufacturing and Industrial Management; “Gheorghe Asachi” Technical University of Iasi, 700050 Iasi, Romania

**Keywords:** AC asynchronous induction motor, sensor, rotary machine, condition monitoring, electric power parameters, signal processing

## Abstract

This paper presents some advances in condition monitoring for rotary machines (particularly for a lathe headstock gearbox) running idle with a constant speed, based on the behaviour of a driving three-phase AC asynchronous induction motor used as a sensor of the mechanical power via the absorbed electrical power. The majority of the variable phenomena involved in this condition monitoring are periodical (machines having rotary parts) and should be mechanically supplied through a variable electrical power absorbed by a motor with periodical components (having frequencies equal to the rotational frequency of the machine parts). The paper proposes some signal processing and analysis methods for the variable part of the absorbed electrical power (or its constituents: active and instantaneous power, instantaneous current, power factor, etc.) in order to achieve a description of these periodical constituents, each one often described as a sum of sinusoidal components with a fundamental and some harmonics. In testing these methods, the paper confirms the hypothesis that the evolution of the electrical power (instantaneous and active) has a predominantly deterministic character. Two main signal analysis methods were used, with good, comparable results: the fast Fourier transform of short and long signal sequences (for the frequency domain) and the curve fitting estimation (in the time domain). The determination of the amplitude, frequency and phase at origin of time for each of these components helps to describe the condition (normal or abnormal) of the machine parts. Several achievements confirm the viability of this study: a characterization of a flat driving belt condition and a beating power phenomenon generated by two rotary shafts inside the gearbox. For comparison purposes, the same signal analysis methods were applied to describe the evolution of the vibration signal and the instantaneous angular speed signal at the gearbox output spindle. Many similarities in behaviour among certain mechanical parts (including their electrical power, vibration and instantaneous angular speed) were highlighted.

## 1. Introduction

Any asynchronous AC induction motor electrically supplied by a three-phase system that drives a rotary machine works primarily as a conversion system for changing electrical power into mechanical power. The consumption of mechanical power is precisely mirrored by the absorption of electrical power from the electrical supply system (with *η* < 1 as the ratio of active mechanical/active electrical power describing the power efficiency). All of the phenomena inside the motor, and especially those either within the driven rotary machine or related to it by a working process powered by the rotary machine and characterized by the variable consumption of mechanical power, should have a description in the evolution of the electrical power parameters measured at the electrical input of the motor. These parameters are mainly the instantaneous current and voltage, the instantaneous electrical power, active electrical power, power factor, reactive power, energy, etc. If these parameters are measurable, this means that any electrical motor (particularly an AC induction motor) has a secondary function as a mechanical power sensor or as a loading sensor and can be useful in condition monitoring, based on the information that flows through the motor from the driven machine (as mechanical power) to electrical system (as electrical power). It is obvious that, when a driving AC induction motor is used as a sensor, the majority of the mechanical phenomena considered for the condition monitoring of rotary machines running in idle or with a working process are better described by the evolution of the electrical power in comparison with the evolution of the electric current, contrary to the present best practices according to the literature.

With 1/*η* being the ratio of active electrical/active mechanical power, and *η* < 1, it turns out that a slight amplification increases the sensitivity of the motor used as a sensor (in other words, a variation in the mechanical power is mirrored by a variation amplified by 1/*η* in the electrical power).

The indirect measurement of mechanical power based on the measurement of electrical power parameters offers several important advantages. First, it is based on a simple computer-assisted experimental setup with measurement devices placed on the electrical supply system of the AC induction motor. The measurement of instantaneous current and voltage acquisition is performed utilizing this setup [1,2,3,4], which can be easily used for any other AC induction motor. This indirect measurement has a second important advantage: it makes it possible to avoid a much more complicated setup for mechanical power measurement that requires a torque sensor and an instantaneous angular speed sensor to be placed on the AC induction motor rotor. However, the behaviour as sensor of an AC induction motor as related to the evolution of the electrical power parameters should be considered as being slightly affected in a negative way by the mechanical dynamics of the rotor, by the variation of the power efficiency *η*, which is not constant, but depends on loading, and by motor slip.

The ability of AC induction motors to serve as mechanical power sensors based on the evolution of the active electrical power is known and is usually exploited in machine tools for: working processes monitoring (e.g., anomalies detection or faults detection on CNC machining processes [5,6,7], cutting power measurement in milling [8,9,10], drilling [11] or turning [12], peak power reduction [13]), tool condition monitoring [14,15,16,17], feed drive systems monitoring [18] or auxiliary systems monitoring [19]. Some researchers have focused on the monitoring [1,2] and optimization [20] of transient regimes on machine tools (acceleration-deceleration) using the evolution of the active electrical power. An approach to the condition monitoring of motor-operated valves was described in [21].

An alternative electric power parameter frequently used to describe the mechanical loading is the active electric energy [22,23,24] as a rate of active electric power, the power being the energy divided by time. Here, increasing the energy efficiency of machine tools [25,26,27] or impact working machines [28] is a research priority.

Because the faults of the rotary parts in rotary driven machines induce periodical mechanical loadings, in addition to the well-known resources related to vibrations, the literature indicates the availability of the instantaneous electric current delivered to the motors as a basis for condition monitoring and diagnosis in time domain and especially in frequency domain, e.g., for gearboxes [29,30,31,32]. More frequently, the instantaneous electric current is used for the diagnosis of induction rotor motors or generators [33,34,35,36]. Unexpectedly, one promising approach in this research is still unexploited: the mechanical loading introduced by a rotary machine driven by an AC induction motor is considerably better described by the evolution of active electrical power or instantaneous active electrical power in comparison with the evolution of instantaneous current, because the definition of the active electrical power involves the instantaneous current, the instantaneous voltage and—essentially—the shift of phase between them, or the power factor, as well. The instantaneous current is not strictly proportional to the instantaneous electrical power. Also, it is well known that the instantaneous current absorbed by an AC induction motor cannot describe some transient regimes, e.g., a regime characterized by negative mechanical power absorption [1].

The few papers on this topic focus mainly on electric motor condition diagnosis based on instantaneous electric power, for stator turn fault detection [37] and mechanical imbalances of the rotor [38], or based on power factor [39] evolution. It seems that, according to the literature, instantaneous electrical power is a better approach than active electric power to the description of free and forced mechanical oscillations, as periodical mechanical loading, in driven machines and systems [40], to the description of simulated vibrations [41] and experimentally revealed vibrations [42] at resonance or to the description of active damping [43] in electrically actuated mechanical systems.

The main objective of this work is to highlight the availability of the evolution for some parameters of electric power absorbed by a three-phase AC asynchronous induction motor—especially instantaneous power and instantaneous active power—in the condition monitoring of a rotary driven machine, particularly a lathe headstock gearbox running idle with a constant rotational speed at the output spindle. The rotary mechanical parts placed inside the gearbox—in normal and abnormal conditions—need to be powered with mechanical power provided by motor. The variable part of this power, which is appropriately considered to be preponderantly a sum of sinusoids, is analysed in electrical equivalent in the time domain and in the low frequency domain in order to find the mathematical descriptions for the sinusoidal components related to the condition of each of these mechanical parts, especially flat belts and shafts in this paper. For comparison purposes, in order to increase confidence in the results, some supplementary resources in condition monitoring offered by the evolutions of gearbox vibration and instantaneous angular speed of the output spindle are also exploited. 

The achievements of this work focus especially on the processing and analysis of electrical power and its components in order to reveal the condition of some rotary parts inside a lathe headstock gearbox running with constant speed, due to the fact that the components of the electrical power are preponderantly deterministic signals. Some interesting phenomena inside the gearbox were revealed. Firstly, a beating vibration phenomenon supplied by a beating instantaneous power phenomenon inside the gearbox, generated by two shafts with imbalanced masses, was highlighted. Especially on this topic, contrary to the expectations, it is proved that the vibration signal is not able to produce a complete description of some of the phenomena involved in condition monitoring. Secondly, the periodical variation of power produced by a flat belt in different working conditions was revealed and investigated.

The rest of this paper is organized as follows: Section 2 presents the computer-assisted experimental setup, Section 3 presents a theoretical approach to the numerical definition of some electric power parameters, Section 4 presents the experimental results and discussion, Section 5 is dedicated to discussion and Section 6 presents the conclusions and the future work.

## 2. The Experimental Setup

The experimental setup is briefly described in Figure 1. A three-phase AC asynchronous induction motor (5.5 KW and 1500 rpm synchronous speed, with two magnetic poles, a delta connection on stator windings and a squirrel-cage rotor placed at the input of a Romanian SNA 360 lathe headstock gearbox) is supplied with a three-phase four-wire system. A star connection on stator windings is used temporarily for motor starting.

The signals delivered to a PC USB oscilloscope (PicoScope 4424 from Pico Technology, Cambridgeshire, UK [44]) by a voltage transformer VT and a current transformer CT (both placed on phase A as sensors) are used by a computer to describe and to analyse the evolution of some electric power parameters (e.g., instantaneous current and voltage, instantaneous electrical power, instantaneous electrical active power, power factor, etc.). The signal delivered by a vibration sensor VS (self-generating velocity detector Geo Space GS 11D, now HGS Products HG4, 5 Hz natural frequency [45]) placed on the lathe gearbox headstock near the jaw chuck is used to describe the vibrations in the horizontal direction, perpendicularly on the spindle axis (also through the medium of the PC USB oscilloscope and the computer).

An instantaneous angular speed sensor (IASS, as a self-generated speed signal sensor, proposed in a previous work [46]) is placed in the jaw chuck of the spindle.

It is expected that some phenomena related with the lathe headstock condition are similarly mirrored in the evolution of the electric power parameters, gearbox vibrations and instantaneous angular speed of spindle. 

According to the gearing diagram from Figure 2, a particular configuration of the lathe gearbox was chosen in order to obtain a theoretical rotational speed of 1052.18 rpm at the spindle. Inside the gearbox, there are different mechanical parts: two flat belt transmissions, spur and helical gears, three friction electromagnetic clutches, three shafts and a spindle mounted on bearings. Nine different manually and electrically selectable speeds are available. The real values of rotational speed experimentally measured for the spindle, shafts and belts are written with red fonts in the gearing diagram from Figure 2.

Each transformer (VT, CT) and each sensor (VS, IASS) delivers an AC signal (as VT_S_, CT_S_, VS_S_ and IASS_S_ signals). Figure 3 describes a short sequence in which these four AC signals are simultaneously generated when the lathe headstock gearbox runs idle (with gearing diagram described in Figure 2).

The VT_S_ and CT_S_ signals theoretically have a 50 Hz frequency. A time delay between the VT_S_ and CT_S_ signals occurs due the phase shift (this happens because the motor stator winding acts as an inductive circuit). This phase shift is variable; it depends mainly on the mechanical loading of the AC induction motor.

The IASS_S_ signal appears [46] as an AC signal with 50 periods on each spindle rotation (nearly 868.5 Hz frequency). The VS_S_ signal contains a major dominant quasi sinus- oidal wave component due to the excited vibration of the entire gearbox on its foundation.

The excitation is generated by the rotation of two unbalanced bodies, a shaft and the output spindle (having a rotation frequency of nearly 17.37 Hz) and slightly amplified by a resonance phenomenon.

## 3. The Numerical Description of Some Electric Power Parameters

The evolution of the electric power parameters of the AC induction motor depends exclusively on the evolution of the instantaneous current and voltage (and the phase shift between).

According to Figure 1, for an AC asynchronous induction motor electrically powered with a 50 Hz symmetrical three-phase (A, B, C) four-wire supply system (N being the neutral wire), the signal CT_S_ delivered by the current transformer CT [2] is proportional with the instantaneous current *i_A_*(*t*), and the signal VT_S_ delivered by the voltage transformer VT is proportional with the instantaneous voltage *u_A_*(*t*) on a single phase (A), both having approximately sinusoidal evolutions with the same period *T* = 1/50 s and a shift of phase *φ_A_* between them. It is supposed that, in a balanced sinusoidal electric supply system, the evolutions of the instantaneous current (IC) and the instantaneous voltages (IV) are the same in all three phases, except for a phase shift angle of 2π/3 radians between phases. Two numerical samples *i_A_*[*t_k_*] (for IC) and *u_A_*[*t_k_*] (for IV) (described by computer, with *i_A_*[*t_k_*] and *u_A_*[*t_k_*] proportional with the numerical description of CT_S_ and VT_S_ signals delivered by the PC USB oscilloscope) give a definition of a numerical sample *p_A_*[*t_k_*] = *u_A_*[*t_k_*]⋯*i_A_*[*t_k_*] of the instantaneous electrical power *p_A_*(*t*) = *u_A_*(*t*)·*i_A_*(*t*) on phase A (as IP_A_), with Δ*t* = *t_k_* − *t_k_*
_− 1_ as sampling period and with *F_s_* = 1/Δ*t* being the sampling frequency. If *i_A_*(*t*) and *u_A_*(*t*) are pure sinusoidal signals, then IP_A_ has a periodical evolution with a period equal with *T*/2 (or 2/*T* frequency as well). Otherwise, IP_A_ has a dominant periodical component with a period equal with *T*/2 (as fundamental) and some harmonics. A part of this, instantaneous electrical power (as instantaneous electrical active power, IAP_A_) is involved in the conversion and the transfer of energy strictly in one direction: from electrical energy absorbed by the AC induction motor to mechanical energy delivered by the motor to the driven rotary machine.

If there are *n* samples *p_A_*[*t_k_*] on each IP_A_ period *T*/2 (with *n·*Δ*t* = *T*/2), then there is a first known approach to estimating a sample *P_A_*_1_[*t_l_*] of IAP_A_ (as IAP_A1_) as the average value of IP_A_ on the *l*th period, as follows:(1)$$P_{A1}\left[t_{l}\right]=\frac{1}{n}\overset{n\left(l+1\right)}{\underset{k=nl+1}{\sum }}p_{A}\left[t_{k}\right]\;\;$$

The time *t_l_* can be described as *t_l_* = *l T*/2 + *T*/4. The sampling period of IAP_A1_ is *T*/2 >> Δ*t.* The sampling rate of IAP_A1_ is *F_s_* = 2/*T*, the Nyquist frequency (or Nyquist limit, involved in the fast Fourier transform currently used in this paper) is *F_Nyquist_* = *F_s_*/2 = 1/*T*. For a 50 Hz frequency of instantaneous voltage, or *T* = 20 ms, the *F_s_* = 100 s^−1^ and *F_Nyquist_* = 50 Hz.

There is a second approach to calculating a sample *P_A_*_2_[*t_k_*] of IAP_A_ (as IAP_A2_), which involves completely removing the dominant component from IP_A_ (2/*T* Hz frequency) and its harmonics (not involved in condition monitoring) using successively narrow band-stop notch filters with middle frequencies at 2/*T* Hz, 4/*T* Hz, 6/*T* Hz and so on. A simpler way [43] is to use a numerical moving average filter with *n* samples in the average (*n* being the integer part of the ratio *T*/(2·Δ*t*)) having first notch frequency at 2/*T* Hz, as follows:(2)$$P_{A2}\left[t_{k}\right]=\frac{1}{n}\overset{n}{\underset{j=1}{\sum }}p_{A}\left[t_{k-j}\right]$$

This filter has also notch frequencies at 4/*T* Hz, 6/*T* Hz, 8/*T* Hz and so on; consequently, it also completely removes all of the harmonics of the dominant component of IP_A_ (200 Hz, 300 Hz, 400 Hz and so on). The IAP_A2_ description has a big advantage compared with IAP_A1_: it keeps the same sampling period Δ*t* as IP_A_ (for *F_s_* = 1/Δ*t*, *F_Nyquist_* = 1/(2Δ*t*)). However, the moving average filter has also a relatively big disadvantage: the IAP_A2_ description contains sinusoidal components, and the amplitudes of these components are strongly attenuated (except those with low frequencies, much lower than 2/*T* Hz).

It is obviously that, in order to describe the total instantaneous active power absorbed by the AC induction motor through all three phases (as IAP), in each sample of IAP (*P*_1_ and *P*_2_) a multiplication factor 3 should be used (*P*_1_ = 3·*P_A_*_1_ and *P*_2_ = 3·*P_A_*_2_).

An important electric power parameter useful in condition monitoring is the instantaneous power factor (IPF) [39], theoretically the same on all phases, as the cosine of the phase shift *φ_A_* (or cos(*φ_A_*)) between instantaneous voltage *u_A_*(*t*) and current *i_A_*(*t*). The time delay *t_dp_* between two successive zero-crossing moments of instantaneous voltage and current (experimentally revealed in Figure 3 by the time delay between VT_S_ and CT_S_ signals) is involved in the definition of the *p*th sample of IPF (as *PF*[*t_p_*]) as follows:(3)$$PF\left[t_{p}\right]=\cos\left(2\pi \frac{t_{dp}}{T}\right)\;\;\;$$

The time *t_p_* can be described as *t_p_* = *p·T*/2 + *t_dp_*/2. Similarly to that of IAP_A1_, the sampling period of IPF is *T*/2 >> Δ*t* (because on each period *T* there are two zero-crossing moments for *u_A_*(*t*) and two zero-crossing moments for *i_A_*(*t*)). A high precision method for detecting the zero-crossing moments for sinusoidal signals was fully described in a previous work [46].

Some other electric power parameters are also available: the active power AP (as an average of IAP on a time interval), the active energy AE (as the definite integral of AP on a time interval), the instantaneous reactive power IRP (the result of multiplication between the samples of IAP and tan(*φ_A_*)), etc.

The numerical description of these parameters is suitable for signal processing in the condition monitoring of the proposed rotary driven machine, e.g., by analysis in the frequency domain.

## 4. Experimental Results

With the lathe headstock gearbox running idle and having the gearing diagram depicted in Figure 2, the evolution of instantaneous current and voltage during 100 s was acquired (with *F_s_* = 25,000 s^−1^ as sampling rate and the number of samples *N_s_* = 2,500,000 or *N_s_* = 2.5 Ms). The evolutions of IP_A_, IAP_A1_, IAP_A2_ and AP_A_ were obtained (with AP_A_ being the active power on phase A, as the average of IP_A_, a sample on each second). A comparison between the simultaneous experimental evolutions of these four electrical power parameters is conducted in a short time sequence in Figure 4 (0.1 s, with 2500 samples on IP_A_ or IAP_A2_, 10 samples on IAP_A1_ and 0.1 samples on AP_A_).

The evolution of AP_A_ (as the very low frequency part of IP_A_, IAP_A1_ and IAP_A2_) indicates the total consumption of the AC motor, mainly as an equivalent of the total amount of active mechanical power necessary to rotate the headstock lathe gearbox running in idle regime. This mechanical power is finally lost by dry and viscous friction.

### 4.1. Some Resources of Signals Processing in Frequency Domain Revealed by Fast Fourier Transform

For gearbox condition monitoring purposes, it is appropriate to use the evolution of the variable parts of IP_A_, IAP_A1_ or IAP_A2,_ which are supposed to contain periodical components. The easiest way to explore these variable parts is to use the evolutions in the frequency domain obtained by the fast Fourier transform (FFT) with Matlab. Figure 5 describes a part of the FFT spectrum evolution (0 ÷ 55 Hz) of IP_A_, IAP_A1_ and IAP_A2_ during the same experiment as before (the lathe headstock gearbox running idle), with IAP_A2_ (IAP_A1_) being shifted vertically by 1.5 W (3 W) and horizontally by 1.5 Hz (3 Hz), respectively, in comparison with IP_A_ (in order to avoid the graphical overlap).

It is obvious that these three evolutions involved in the description of electric power are deterministic signals (each peak is related to a sinusoidal component of the electric power); these three FFT spectra are very similar. All of the spectra have (the same) high resolution in frequency (*F_s_*/*N_s_* = 0.01 Hz) with the same sampling rate (*F_s_* = 25,000 s^−1^) for IP and IAP_A2_ (with *N_s_* = 2.5 Ms), while the IAP_A1_ spectrum has the sampling rate *F_s_*/*n* =100 s^−1^ and *N_s_*/*n* = 10,000 samples. Because of a low Nyquist limit, the spectrum of IAP_A1_ ends at 50 Hz (*F_s_*/(2*n*) = 50 Hz), while, for IP_A_ and IAP_A2,_ the spectra end at *F_s_*/(2) = 12,500 Hz.

A zoomed-in detail in the area marked with A in Figure 5 is described in Figure 6.

Figure 7 describes the first major peak from Figure 6 (approx. 27.5 Hz frequency) on all three types of powers (not shifted). It is obviously that—related to amplitudes—the best description of these sinusoidal components is given by the FFT of IP_A_ evolution (compared with IAP_A1_ and IAP_A2_). 

As major disadvantages, the evolution of IAP_A1_ has a low sampling rate and a low Nyquist limit because of its definition as an average (Equation (1)), while the amplitudes of the FFT peaks on the IAP_A2_ spectrum strongly decrease with frequency because of filtering (Equation (2)), as is clearly indicated in Figure 8 (a sequence of spectra between 50 ÷ 105 Hz).

Contrary to expectations, here the fundamental of the dominant component from IP_A_ (100 Hz theoretical frequency and 99.97 Hz real frequency and 923.2 W in amplitude) is not completely eliminated by the moving average filter on IAP_A2_ (there is still a peak of 4.012 W in amplitude). This happens because the spectrum is calculated by averaging a long sequence (100 s). During this sequence, the frequency of the supply of instantaneous voltage (and current as well) varies slightly around 50 Hz, as we will experimentally prove later on, in Section 5. This also implies a variation in the frequency of the dominant component in IP_A_, leading to a slight decrease in the filtering efficiency.

There are two reasons to prefer IP_A_ evolution for condition monitoring: firstly, it is easy to obtain the description of its evolution in time; secondly, it provides the best description of the sinusoidal components in the FFT spectrum.

In the FFT spectrum of IP_A_, the measurement unit for the variable components involved in condition monitoring can be considered to be the watt (due to the similarities of the FFT amplitudes with IAP_A1_ and IAP_A2_ already revealed). For a complete description of the periodical phenomena inside the lathe headstock gearbox as mirrored in the electric power evolution, it is suitable to use the total instantaneous power *p*(*t*) absorbed by the AC motor (or IP, as being three times bigger than IP_A_, with *p*(*t*) = 3*·p_A_*(*t*)). 

These periodical phenomena revealed before (as sums of sinusoids) can be associated with the normal behaviour (or malfunction) of gearbox mechanical parts (shafts, belts, bearings, etc.) using the relationship between the peak frequencies of fundamentals in the FFT spectrum and the rotation frequency of these parts inside the gearbox (running idle with constant rotation speed). Figure 9 and Table 1 describe the FFT spectrum of IP as having sinusoidal components (the fundamentals and some harmonics, in a frequency range of 0 ÷ 40 Hz) generated by the mechanical parts of the gearbox running idle, for the gearing diagram described in Figure 2. This spectrum, with the origins of its main peaks described in Table 1, is the first argument that IP is a predominantly deterministic signal, well distinguished from the noise. 

The average value of the rotation frequency of shaft III and the spindle (17.37 s^−1^, or 1042.2 r.p.m., measured using the IASS sensor [46]) is exactly the value *f_E_* of the peak E. This means that these two mechanical parts (having theoretically the same frequency of rotation) produce together, for certain reasons, a mechanical loading (with the variable part revealed in Figure 9 and Table 1 by the spectrum component E as the fundamental and E_1_ as the first harmonic) on each rotation. This issue will be discussed later. As expected, because of the gears (and gear ratios), there are kinematic relationships between the frequencies of fundamentals *f_D_*, *f_E_* and *f_F_* (*f_D_* being the rotation frequency of the shaft II, *f_F_* the rotation frequency of the shaft I) as follows: *f_D_* = *f_E_·*38/48 and *f_F_* = *f_D_*·58/37. The ratio of flat belt 1 transmission ((139 + 2)/(121 + 2)) implies a well accomplished demonstration of the relationship *f_G_* = *f_F_*·(139 + 2)/(121 + 2). Here the thickness of the flat belt is 2 mm.

It is interesting to know why and how flat belt 1 (Figure 10) introduces a periodical instantaneous power (described in Table 1 by the components with frequencies *f_A_, f_A_*_1_ ÷ *f_A_*_6_). It is probably a matter of the stiffness variation of the belt along its entire length (*l_fb_*_1_). Flat belt 1 continuously moves with *f_A_* cycles per second. 

When a portion with low stiffness (due to belt manufacturing errors or due to wear, such as a crevice or a tear) is placed on the tight side of the belt transmission, the transferred mechanical power decreases. In contrast, when this portion is not placed on the tight side, the transferred mechanical power increases. 

This means that a torsional vibration is generated in the power transmission (having *f_A_* as its fundamental frequency, and *f_A_*_1_ ÷ *f_A_*_6_ as harmonics). If *D_d_*_1_ is the diameter of the driven pulley, then the frequency *f_A_* of the fundamental component introduced by the first belt in the IP evolution is simply calculated as *f_A_* = *f_F_·l_fb_*_1_/(π*·D_d_*_1_).

Similar comments are available for flat belt 2 (here the frequencies *f_B_, f_B_*_1_ ÷ *f_B_*_3_ are involved). In contrast with the first belt, flat belt 2 is 35 years older; it has bigger amplitudes for the fundamental and 2nd harmonic (on the *f_B_* and *f_B_*_2_ frequencies). Certainly, using the FFT of IP, structural damage to the belts can be detected and described (condition monitoring) long before they break by a severe increase in the amplitude of the fundamentals and/or some harmonics. This issue is supplementarily addressed in Section 4.2.

The amplitude of all other components (peaks) from FFT spectrum of IP revealed in Figure 9 and Table 1 are useful for condition monitoring of the shafts and the output spindle. 

The behaviour of the gearbox parts can also be described by IPF spectrum, according to Figure 11, also, an FFT spectrum sequence between 0 ÷ 40 Hz. Almost all components already revealed in the FFT spectrum of the IP evolution (Figure 9) are also well described in the FFT spectrum of IPF (except B_3_). This means that the resources of the IPF spectrum are also useful in experimental research on condition monitoring. However, as a major disadvantage, the Nyquist limit for IPF evolution is very low: only 50 Hz.

Apparently, increasing the number of samples *N_s_* of the IP sequence (e.g., from *N_s_* = 2.5 Ms to *N_s_* = 5 Ms, with the same sampling rate *F_s_* = 25,000 s^−1^) decreases the frequency resolution of the FFT spectrum (from 0.01 to 0.005 Hz) while yielding better descriptions of the components in the frequency domain. Unfortunately, because of a slow fluctuation in the rotational speed of the AC induction motor rotor (mainly because of motor slip with the fluctuation of mechanical loading), an undesirable phenomenon occurs: spectral smearing [47]. As result, the FFT components are poorly described, as Figure 12 indicates (a zoomed-in detail, in the area of peaks D_1_ and B_2_ as revealed in Figure 9).

It is interesting for condition monitoring that the periodical phenomena earlier revealed in Figure 9 and Figure 10 are also mirrored in the FFT spectrum of the IAS measured at the spindle, as Figure 13 indicates. The IAS evolution also depends on mechanical loading, which shows another use for the AC induction motor as a sensor, as already revealed in a previous work [46]. The average value of the IAS of the spindle is 109.14 rad·s^−1^ (with 17.37 Hz average rotational frequency), the sampling frequency is 1737 s^−1^ and the number of samples is 173,700.

For condition monitoring purposes (and better results), it would be much better if the IAS sensor (IASS) were placed on the rotor of the AC induction motor (because normally the jaw chuck of the spindle should be permanently available to the work piece during the working process).

The FFT spectrum of the vibration signal VS_S_ (from Figure 14, proportional with vibration velocity) delivered by the vibration sensor VS placed on the gearbox (Figure 1) describes the same periodical components already found before in the IP, IPF and IAS evolutions in frequency domains (Figure 9, Figure 10 and Figure 13).

In Figure 14, the component E is dominant (as already was stated in Figure 3); it has the highest amplitude in the spectrum (158.52 mV, on the spindle rotational frequency of 17.37 Hz). In brief, this vibration component E is generated by shaft III and the output spindle rotation having unbalanced masses. It is strongly amplified by a mechanical resonance due to a low damped vibration mode of the entire lathe placed on its foundation (18.79 Hz frequency, 5.899 s^−1^ damping constant [48]). This issue will be supplementarily discussed later on. Apart from the comparison of the similarities between Figure 9 and Figure 14, giving two different ways to describe the behaviour of the same gearbox parts (with IP and VS_S_ spectra), it is more appropriate to check these similarities using the pointwise product of FFT spectra for IP and VS_S_, as Figure 15 indicates. This is possible because the evolutions of IP and VS_S_ are numerically described with the same sampling rate (*F_s_* = 25,000 s^−1^) and the same number of samples (*N_s_* = 2.5 Ms).

It astonishes us that, contrary to any expectations, there is not any similarity between the FFT spectra of IC (depicted in Figure 16) and IP (from Figure 9). It seems that the IC evolution is not suitable for condition monitoring, at least when using our method to obtain the FFT spectra.

However, we discovered that if, instead of IC spectrum, the full-wave rectified instantaneous current (as RIC) spectrum is used, the result is totally changed, as Figure 17 indicates (the same magnitude was used for Figure 16 and Figure 17). Now, there are strong similarities between the FFT spectra of RIC and IP (Figure 9 and Figure 17 have almost the same components, which are placed at the same frequencies in spectra). We should mention that, while IC is described with *i_A_*[*t_k_*] samples, RIC is described with |*i_A_*[*t_k_*]| samples (as a modulus function applied to *i_A_*[*t_k_*] sample). It is obvious that RIC also provides useful information for condition monitoring, but probably no better than IP, mainly because IP (and its derivatives IAP_A1_ and IAP_A2,_ as well) is more strongly related to mechanical loading than IC and RIC. It is known that IC and RIC are not able to describe a negative mechanical power flow delivered by a motor during a transient regime [1], when the AC motor works as a braking system.

A similar study was performed on IV, the second constituent of IP. Theoretically, IV should be permanently constant (as amplitude and frequency), indifferent to the IP absorbed by the AC induction motor. However, because each of the three wires used to supply the AC motor has a non-zero electrical resistivity, the variation of IC generates a variation of the voltage drop in these wires (due to the Ohm’s law), which implies the variation of IV measured at the input of the AC induction motor. 

As expected (and similarly to the IC spectrum), the IV spectrum (depicted in Figure 18) does not have any similarities with the IP spectrum from Figure 9. However, if the IV evolution (described with *u_A_*[*t_k_*] samples) is replaced by the full-wave rectified instantaneous voltage (as RIV, described with |*u_A_*[*t_k_*]| samples), the spectrum of RIV (depicted in Figure 19) has strong similarities with the IP spectrum (and the RIC spectrum as well), as a comparison between Figure 9 and Figure 19 proves. This means that RIV also provides useful information for condition monitoring, but certainly no better than IP or RIC (due to an important noise in the spectrum).

For the time being, we do not have a feasible explanation for why the IC and IV spectra (generated by FFT in Matlab) are not relevant in condition monitoring, in comparison with the RIC and RIV spectra. However, we should mention that there are strong similarities between the IC and IV spectra, related to the frequency of some relevant components, as a comparison between Figure 16 and Figure 18 proves.

It is not surprising that the behaviour of many of the gearbox’s rotating parts is mirrored simultaneously, particularly in FFT spectra of the IP and VS_S_ signals. As an example, any unbalanced rotating shaft inside the gearbox acts as a sinusoidal vibration exciter that moves the lathe periodically on its foundation (acting as an excited spring-mass-damper system). A fundamental on-shaft rotation frequency on the FFT spectrum of VS_S_ occurs.

When we talk about the amplitudes of the vibratory motions generated by the unbalancing of shaft III and the output spindle, these amplitudes depend on the mass unbalanced, the almost identical rotation frequency (17.37 Hz) and the proximity to the mechanical resonance frequency of the lathe on its foundation (a 18.79 Hz frequency, with a damping constant of 5.899 s^−1^ [48]).

Obviously, the mechanical power necessary to supply this sinusoidal motion (as vibration excitation power, maximal at mechanical resonance) is delivered by the AC induction motor. This excitation power is a constituent of IP absorbed by the motor from the electrical supply system and is described as a peak in the IP FFT spectrum. As an example of gearbox failure, a bent shaft regularly needs a mechanical power to rotate it, such as drive power. Corresponding to the variable (periodical) part of this drive power, a fundamental on its rotation frequency and some harmonics on the IP FFT spectrum are generated. The rotation of this bent shaft generates a deformation of the gearbox housing, sometimes perceived as vibration by the vibration sensor (a fundamental on its rotation frequency and some harmonics on FFT spectrum of VS_S_ are generated). A similar approach is available for many other anomalies involving the rotating parts inside the gearbox (e.g., bearing misalignment, gear run out, damaged belts, etc.).

Sometimes, the mechanical variable power necessary to rotate a bent shaft with a mechanical imbalance (e.g., shaft III or/and the output spindle, with the variable power depicted by peak E in Figure 9) has two cumulated constituents with the same (rotation) frequency, often with a phase shift between: one constituent is the periodical part of the drive power used to rotate the shaft if the imbalance is ignored; the other constituent is the vibration excitation power, used to supply only the vibration motion of the lathe on its foundation, excited by the mechanical imbalance of this rotary shaft.

A particular rotational property of shaft III and the output spindle, revealed in a previous work [48], is useful to highlight the existence of these two constituents in the evolution of IP (drive power and vibration excitation power): a beating phenomenon in the vibrations of the gearbox (together with the lathe on its foundation) with nodes and antinodes, as is experimentally described by the evolution of the VS_S_ signal (curve 1 in Figure 20) with *N_s_* = 5 Ms and *F_s_* = 25,000 Hz during 200 s, with the gearbox running idle as previously.

This beating phenomenon (generated by vibration superposition with constructive and destructive interferences) occurs because there is a very small difference (less than 0.01 Hz) between the rotation frequencies (or IAS) of the output spindle and shaft III (due to a slight sliding of flat belt 2 on its pulleys) and due to a mechanical imbalance of shaft III and the output spindle. The amplitude of beating vibrations increases a lot because of a resonant amplification of the lathe suspension on its foundation, previously highlighted in [48]. Normally, we should observe two very close peaks on the FFT spectrum of VS_S_: a peak generated by shaft III and another peak generated by the output spindle. Nevertheless, because there is not a small enough frequency resolution in the FFT of VS_S_, and because there is a very small difference in the rotation frequency of the output spindle compared with that of shaft III, a single peak occurs in the VS_S_ spectrum (as dominant peak E in Figure 14, having the frequency *f_E_* = 17.37 Hz). 

The variation in time of the amplitude of the dominant peak E (and the relationship with the amplitude of the beating vibration phenomenon) should be experimentally revealed. This paper proposes an appropriate method, as described below.

It is expected that the FFT spectrum of a short movable sequence of VS_S_ (as SMSVs, with *N_ss_* samples, and *N_ss_* << *N_s_*) will reveal that the amplitude of peak E (the dominant component in VS_S_) is variable in time, with an evolution close to that of the amplitude of the beating vibration. On the FFT spectrum of each SMSVs having the average time *t_AEj_,* a value for the amplitude of peak E (as AE_Vs_) is available. Any two successive SMSVs (eventually with an overlap between them) produce two successive samples, *AE_Vs_*(*t_AEj_*) and *AE_Vs_*(*t_AEj_*
_+ 1_). 

There is a major difficulty to this approach: the FFT frequency resolution of SMSVs is very small (*F_s_*/*N_ss_* >> *F_s_*/*N_s_*). In order to detect exactly the (average) value of the amplitude of the peak E inside a SMSVs, it is mandatory to accomplish exactly the condition *k·F_s_*/*N_ss_* = *f_E_*. In other words, the frequency *k·F_s_*/*N_ss_* of the *k*th sample on the FFT spectrum of each SMSVs should have exactly the value *f_E_*, or should at least have the closest value possible of *k·F_s_*/*N_ss_* relative to *f_E,_* taking into account the fact that *k* and *N_ss_* are positive integers. For *f_E_* = 17.37 Hz and *F_s_* = 25,000 s^−1^, a computer-assisted trial algorithm produces the appropriate value of *N_ss_* = 27,346 (for *k* = 19), with *k·F_s_*/*N_ss_* = 17.369999 Hz ≈ *f_E_* = 17.37 Hz.

In Figure 21, the advantages of this method of using the FFT spectrum of SMSVs are revealed.

Curve 1 depicts the peak E on the entire FFT spectrum of VS_S_, with *N_s_* = 5 Ms. It describes the frequency very well but gives a bad description of amplitude (expected to be close to the average value of the amplitudes on curve 1, Figure 20). Curve 2 in Figure 21 depicts the peak E on the FFT spectrum of a single SMSVs placed in the area of the first antinode in Figure 20, while curve 3 in Figure 21 depicts the peak E on the FFT of a single SMSVs in the area of the second node in Figure 20 (both SMSVs with *N_ss_* = 27,346 samples for the best fulfilment of the previously defined condition *k·F_s_*/*N_ss_* = *f_E_*). Despite a very small frequency resolution, as expected, there is a good description of the frequency and, very importantly, a good description of the amplitudes (compared with the amplitudes of the nodes and antinodes in Figure 20, curve 1). Using this SMSVs movably on the entire VS_S_ signal (*N_s_* = 5 MS, with 1000 samples of overlap between two successive SMSVs), the complete AE_Vs_ evolution is obtained and described with curve 2 in Figure 20, which uses 4946 samples. A computer program was developed in order to find the description of each *AE_Vs_*(*t_AEj_*) sample on the AE_Vs_ evolution.

Two arguments explain why the AE_Vs_ evolution from curve 2 in Figure 20 is slightly different from the amplitude of VS_S_ (of curve 1): firstly, each sample on AE_Vs_ is the average of an SMSVs; secondly, the VS_S_ signal contains many other vibration components (some of them already revealed in Figure 14). A wrong value for *N_ss_* in SMSVs (e.g., *N_ss_* = 21,282) inaccurately describes the peak E in the FFT of SMSVs (in frequency and amplitude), as Figure 21 proves, with curve 4 for the amplitude in the first antinode and curve 5 for the amplitude in the second node. Consequently, the evolution of AE_Vs_ found using this wrong value of *N_ss_* (curve 3 in Figure 20) is not correctly described. 

A hypothesis can be formulated: the beating phenomenon in the vibration should be mandatorily mechanically powered by the electrical power absorbed by the AC motor. A beating phenomenon generated by the superposition of two sinusoidal components of the IP having very close frequencies should take place (one component—having the frequency *f_EIII_*—related to shaft III’s imbalance and its rotational frequency, the other—having the frequency *f_ES_*—related to the output spindle’s imbalance and its rotational frequency), both as mechanical power components used to excite these vibrations. The same type of evolution for amplitude of peak E (already revealed in the VS_S_ evolution, curve 2 in Figure 20) should be observed in the IP evolution (as AE_IP_) during the same experiment previously conducted for VS_S_ (*N_s_* = 5 Ms and *F_s_* = 25,000 Hz). Here, the same FFT analysis with short movable sequences of IP (as SMSIP, also with *N_ss_* = 27,346 samples and 1000 samples of overlap) was performed. The previously formulated hypothesis is fully confirmed in Figure 22.

It is obvious that AE_IP_ (curve 1) has a noisy evolution. It is experimentally proved now that AE_IP_ has relatively important variations of amplitude. In order to reduce the noise, this evolution from curve 1 can be low pass numerically filtered through the medium of three consecutive moving average filters (with 20, 10 and 5 samples in average), with the result depicted with curve 2 (as averaged AE_IP_, also having 4946 samples). A strong reduction of the noise can be obtained also (before filtering) if the number of samples from SMSIP is doubled, tripled, etc., keeping the same overlap. 

The FFT analysis of the IP evolution was repeated in exactly the same manner for the RIC evolution, in order to find out the evolution of the peak E amplitude in RIC (with a short movable sequence of RIC as SMSRIC, also with *N_ss_* = 27,346 samples and 1000 samples overlap). The results (as AE_RIC_ evolutions) are depicted in Figure 23. All the considerations and comments made in Figure 22 (e.g., the filtering of curve 1) are also applicable here. The similarities between curves 2 in Figure 22 and Figure 23 are obvious, despite the fact that curve 2 from Figure 23 (as averaged AE_RIC_ evolution) is more poorly defined.

In the start and in the end areas of Figure 22 and Figure 23 (marked with pink dots), a couple of distorted values generated by filtering (edge effect) should be ignored.

A better comparison between the averaged AE_IP_ and AE_RIC_ evolutions is provided in Figure 24. Here, the averaged AE_RIC_ evolution was graphically fitted (related only by magnitude) against the averaged AE_IP_ evolution in such a way as to highlighted that there is no time delay (or phase shift, either) between these evolutions. As expected, there is no shift of phase between the beating phenomena described with the AE_IP_ or AE_RIC_ averaged evolutions (both describing the variable part of the mechanical loading).

It is obvious also that the averaged AE_IP_ (curve 2 in Figure 22) and AE_Vs_ (curve 2 in Figure 20) have similar evolutions. The beating phenomenon in vibration is supplied by a beating phenomenon in IP (the AC motor supplies with mechanical power the forced vibrations). A better comparison between the averaged AE_IP_ and AE_Vs_ evolutions is provided in Figure 25.

Here, similarly to the averaged AE_RIC_ from Figure 24, the AE_Vs_ evolution was graphically fitted (related only by magnitude) against the averaged AE_IP_ evolution in such a way that only the time delay between them (and the phase shift, as well) is highlighted. 

This phase shift proves that there are two variable IP components used by shaft III (having the same frequency *f_EIII_*): the fundamental of the variable part of the drive power (*p_vdpIII_*, used to rotate the shaft if the imbalance is ignored) and the excitation power (*p_epIII_,* used only to excite the gearbox due to shaft III imbalance during the rotary motion). Similarly, there are two IP components generated by the output spindle (having the same frequency *f_ES_*): the fundamental of the variable part of drive power (*p_vdpS_,* used to rotate the output spindle if the imbalance is ignored) and the excitation power (*p_epS_,* used only to excite the gearbox due to spindle imbalance during the rotary motion). Generally, there is a shift of phase (unknown for the time being) between the *p_vdpIII_* and *p_epIII_* evolutions; also, there is a shift of phase (unknown) between the *p_vdpS_* and *p_epS_* evolutions. A first IP beating phenomenon (as IP_vdp_) takes place through the addition of the powers *p_vdpIII_* + *p_vdpS_* = *p_vdp_* (with *p_vdp_* having a variable amplitude and the frequency *f_E_*). A second IP beating phenomenon (as IP_ep_) takes place through the addition of the powers *p_epIII_* + *p_epS_* = *p_ep_* (with *p_ep_* having a variable amplitude and the frequency *f_E_*). Generally, is expected that, for a certain IAS spindle, there is a constant shift of phase between *p_vdp_* and *p_ep_*.

The IP_ep_ beating phenomenon supplies the vibration beating phenomenon, with no phase shift in between. The IP_vdp_ and IP_ep_ beating phenomena defines (by interference) a third IP beating phenomenon (as IP_E_) by the addition of the powers *p_vdp_* + *p_ep_* = *p_E_,* with the amplitude evolution already described in Figure 22 (curve 1). It has the averaged AE_IP_ as its average amplitude evolution (curve 2 in Figure 22) and the frequency *f_E_*. Since the amplitude of *p_vdp_* is not zero, then *p_E_* ≠ *p_ep_*. As a consequence, there is a time delay (shift of phase) between the averaged AE_IP_ and AE_Vs_, as Figure 25 clearly indicates. The time delay (shift of phase) between the averaged AE_IP_ and AE_Vs_ evolutions can be changed if the amplitude of *p_vdp_* or *p_ep_* (or both) is changed, because IP_E_ depends on IP_vdp_ and IP_ep_ (their sizes and shapes and the phase shift between them).

As a first example, the influence of *p_ep_* on *p_E_* is indirectly proved by a new experiment in almost identical conditions (*f_E_* = 17.39 Hz and *N_ss_* = 29,471), with a 36.8 Kg additional mass (a rectangular parallelepiped made of steel, RPS) placed on the lathe gearbox (Figure 26). 

The circumstances of the excited vibrations (of the lathe on its foundation) are changed: because of the additional mass, the resonant amplification diminishes, and the resonance frequency moves away from the excitation frequency. As Figure 26 proves, there are two major consequences: firstly, the time delay (shift of phase) between the averaged AE_IP_ and AE_Vs_ evolutions decreases significantly; secondly, the peak-to-peak amplitude of *p_E_* decreases from 33.57 W in Figure 25 to 24.4 W here. During this new experiment, the ambient temperature is higher (approximately with 5 degrees Celsius), the sliding of flat belt 2 increases, the difference between the rotational frequencies of shaft III and the output spindle increases, and, consequently, the beat period decreases from 96 s (in Figure 25) to 86.6 s here in Figure 26.

The experiment from Figure 26 was repeated in new conditions. This time, a sheet of rubber 6 mm in thickness was placed between the additional mass and the gearbox. As Figure 27 proves, there is a big change in the time delay between the averaged AE_IP_ and AE_Vs_ evolutions and a noticeable change in the AE_Vs_ average values (from 205.65 mV in Figure 26 to 161.7 mV in Figure 27). The additional mass RPS and the sheet of rubber work together as a passive dynamic vibration damper (PDVD). The amplitude of the average vibration decreases because a part of the instantaneous power *p_ep_* absorbed by the AC induction motor and converted in mechanical power (previously used to supply the gearbox vibration) is now absorbed by the PDVD and converted irreversibly into heat inside the rubber sheet (which also works as a damper).

Under the same experimental conditions as in Figure 25, a short sequence (approximately 17 s in duration) of a longitudinal interrupted turning process with manual feed, on a steel work piece (30 mm diameter) with radial run out (3 mm, bigger than the depth of cut—1.5 mm), acts temporary as a periodical mechanical power consumption phenomenon at the output spindle (as an additional part of the variable drive power *p_vdpS_*). The influence of this sequence on *p_E_* and on the evolution of the averaged AE_IP_ is clearly described in Figure 28, with a local increasing of the averaged AE_IP_ and a shift of phase in comparison with the AE_Vs_. The influence of this cutting process on the gearbox vibrations (more exactly on the AE_Vs_) is null, as the action–reaction periodical cutting forces (against the work piece, respectively, against the cutting tool) cancel each other out. However, the AE_Vs_ decreases temporary when the cutting process ceases (because of the manual removal of the cutting tool from the process, some temporary damping is introduced). 

The results and comments about Figure 25 and especially Figure 28 prove that some important phenomena involved in condition monitoring (e.g., related to a periodical mechanical loading) are not perceptible by a vibration sensor (however it may be useful for other phenomena). Also, it is highlighted here that there are two ways to describe a forced periodical vibration: directly, by using an appropriate vibration sensor, or indirectly, by measuring the mechanical power at vibration excitation source (here through the medium of the IP absorbed by the AC induction motor).

It is interesting that the beating phenomenon generated by shaft III and the spindle (previously illustrated in averaged AE_IP_ and AE_Vs_ evolutions) is also mirrored in the evolution of the amplitude of the peak E on the FFT spectrum of the IAS (as AE_IAS_) evolution (*N_s_* = 347,406 samples, *F_s_* = 1737 s^−1^). This evolution was obtained by the same procedure of analysis on the FFT with short movable sequences of IAS signal (as SMSIAS with *N_ss_* = 2150). Figure 29 presents the AE_IAS_ evolution (AE_IP_ and AE_Vs_, too) during the experiment already described in Figure 25.

Here, the AE_IAS_ evolution has a peak-to-peak variation of 0.0393 rad/s and an average value of 0.601 rad/s. It would be assumed that there is not a significant phase shift between AE_IAS_ and averaged AE_IP_, but this hypothesis is not confirmed by Figure 29. There is a shift of phase generated by the IAS sensor (a stepper motor working as a two-phase 50 pole AC generator [46]). A false periodical IAS component is generated by IASS due to the constructive inaccuracy (as measurement error, with the fundamental component having the frequency *f_ES_*). An IAS beating phenomenon occurs, caused by the addition of three sinusoidal components: a sinusoidal component generated by shaft III with the frequency *f_EIII_* (a consequence of variable power *p_vdpIII_* + *p_epIII_*), a sinusoidal component generated by the output spindle with the frequency *f_ES_* (a consequence of variable power *p_vdpS_* + *p_epS_*), and the fundamental of the IAS measurement error (also sinusoidal), with the frequency *f_ES_*. If the angular position of the IASS rotor related to angular position of spindle is changed, then the phase shift between AE_IAS_ and the averaged AE_IP_ evolution must also change. This is proved by the evolutions from Figure 30 achieved in a new experiment, in similar conditions to those in Figure 29. Here, there is a 0.0326 rad/s peak-to-peak variation and a 1.148 rad/s average value of AE_IAS_, significantly bigger than before.

If the IASS rotor is attached to another shaft (e.g., shaft II, shaft I or the AC motor’s rotor), the measurement error induced by IASS (related to the negative influence on peak E’s amplitude on the FFT spectrum of IAS, in Figure 13) disappears, and the AE_IAS_ and averaged AE_IP_ evolutions should then be in phase.

For condition monitoring purposes concerning other parts of the gearbox (e.g., gears, bearings, etc.), the best approach is to use the entire FFT spectrum of IP (or RIC and eventually RIV as well), mainly in order to check the values and the evolutions of the amplitudes of high order harmonics of the fundamentals generated by the shafts where the gears and the bearings are placed. 

All of the harmonics of the fundamentals on 50 Hz and 100 Hz (electrically generated in the IP or RIC spectra) should be ignored. The availability of the FFT spectrum for this purpose is proved in Figure 31, which depicts a short sequence of the IP spectrum, and in Figure 32, which depicts a short sequence of the RIC spectrum. The same range of frequencies (40 ÷ 140 Hz, as example) is used for both spectra.

It is obvious that there are strong similarities between these IP and RIC spectra (at least related to the frequency of the spectral peaks). It is important to mention that some peaks in the IP and RIC spectra are related to the excited torsional vibration modes of the shafts and spindle, this being a challenge for a future study.

In contrast with IAP_1_, the IP (or RIC, RIV) spectrum does not (invariably) have an upper limit at 50 Hz. The value of the Nyquist frequency for the IP (or RIC, RIV) spectrum (12,500 Hz in Figure 31 and Figure 32) depends only on the value of the sampling frequency (25,000 s^−1^ here). 

It is expected that many other gearbox condition monitoring techniques presently used in vibrations or instantaneous current analysis in the frequency domain are applicable to the IP (or RIC) evolutions.

For non-stationary processes (e.g., starting, stopping and the changing of spindle speed by changing the gearing diagram), or, eventually, for a catastrophic failure, the FFT analysis is useless (except probably the analysis with the FFT of short movable sequences). Nevertheless, these processes can be very well characterized using the evolution of the low pass filtered instantaneous average electrical power IAP_1_, as was proved in [1].

### 4.2. An Approach on IAP_1_ Signal Components Identification by Curve Fitting

The FFT spectrum of IP describes the frequency of each component well and the amplitude fairly well. A better approach to the study of the behaviour of the gearbox parts mirrored in the IP evolution is possible if a complete description of each variable IP component (as a sinusoidal evolution, with amplitude, frequency and phase at origin of time) is achieved by curve fitting (with the sum of sinusoids model) as a signal components identification procedure. For curve fitting, two options were explored, both having similar results: a method proposed by us (slower), and a method based on Curve Fitting Tool (faster), both conducted in Matlab. 

However, at first glance, it is relatively difficult to identify the IP components, because the IP has a high Nyquist frequency and too many sinusoidal components (including the electrically generated dominant on 100 Hz and its harmonics). If we are interested only in the description of low frequency IP sinusoidal components (e.g., between 0 and 40 Hz, as previously), then it is better to use the curve fitting of the IAP_1_ evolution (the instantaneous active power absorbed by the AC motor, defined as being three times IAP_A1_), which has a Nyquist frequency of 50 Hz and a sampling frequency of *F_s_* = 100 s^−1^. Of course, in the range of 0 ÷ 50 Hz, the IP and IAP_1_ have the same sinusoidal components. For an IAP_1_ sequence with a duration of 5 s (*N_s_* = 500 samples), the curve fitting procedures allow the determination of at least 38 of the most significant sinusoidal components (some of them described in Table 2). At each step in the curve fitting method, a single sinusoidal component is identified (usually that with the highest amplitude) and mathematically removed from the IAP_1_ evolution.

The first confirmation of the correctness of this identification is provided in Figure 33. Here, the blue evolution is the FFT spectrum of IAP_1_, while the red evolution is the FFT of the residual evolution after these identified 38 sinusoidal components were removed (mathematically subtracted) from IAP_1_. It is obvious that the FFT of the residual does not contain significant peaks anymore.

Because of a small resolution of the FFT for the IAP_1_ spectrum (only 0.2 Hz in Figure 33, by comparison with the resolution of 0.01 Hz for IP in Figure 9), some variable components in the IAP_1_ spectrum are hidden (e.g., C_1_, A_3_, A_5_, A_6_, E_1_). However, they are well revealed by curve fitting.

Table 2 presents the values of the amplitude and frequency of the 20 most significant sinusoidal components from IP (already revealed before in Table 1) and the values of amplitude, frequency and phase at origin of time for the same sinusoidal components, but as determined by curve fitting of IAP_1_.

The second confirmation of the correctness of the identification and description of low frequency IAP1 sinusoidal components by curve fitting is provided in Figure 34. Here, the experimental evolution in the time domain for the variable part of IAP_1_ is depicted (as a blue curve, a short sequence with duration of 1.5 s, with 150 samples, at the beginning of the sequence analysed by curve fitting), along with the evolution of the variable part of the power theoretically built by the mathematical addition of these 38 sine-identified components (the evolution depicted with a red curve). It is obvious that these two evolutions fit fairly well each other. The difference between these evolutions is described by quite a small residual (the experimental evolution minus the theoretical one) in Figure 34. The results from Figure 33 and Figure 34 (and Table 2 as well) are another argument that IP (IAP_1_) is a predominantly deterministic signal, well distinguished from the noise by a relatively small residual.

By mathematical addition of the sine components A (as fundamental) and A_1_ ÷ A_6_ (as harmonics) identified by curve fitting, it is possible to find the evolution of the variable part of IAP_1_ (or IP as well) generated by flat belt 1. Because, in Table 2, the frequencies *f_Ai_* of the harmonics A_i_ (depicted in the 4th row, identified by curve fitting) are not perfectly related to the frequency *f_A_* of the fundamental (with the relationship *f_Ai_* = (*I* + 1)*·f_A_*), mainly because of errors in curve fitting, it is more appropriate to approximate the value *f_A_* with an average value *f_Aa_* calculated as described in Equation (4).
(4)$$f_{Aa}=\frac{1}{7}\;\left(f_{A}+\overset{6}{\underset{i=1}{\sum }}\frac{f_{Ai}}{\left(i+1\right)}\right)\;\;\;\;\;\;\;$$

In this way, the frequencies of the harmonics are now approximated as *f_Aai_* = (*i* + 1)*f_Aa_* (e.g., *f_A_* = 5.336 Hz becomes *f_Aa_* = 5.3336 Hz, *f_A_*_1_ = 10.664 Hz becomes *f_Aa_*_1_ = 10.6672 Hz, … *f_A_*_6_ = 37.330 Hz becomes *f_Aa_*_6_ = 37.3352 Hz).

In such a way, a more accurate description of the evolution of the variable part of IAP_1_ (or IP as well) generated by flat belt 1 is possible, as curve 1a in Figure 35 indicates.

Here, curve 1a results from a mathematical addition of the sinusoidal components A, A_1_ ÷ A_6_ (having the frequencies *f_Aa_*, *f_Aai_* and the amplitudes indicated in Table 2) with a sampling frequency of 25,000 s^−1^, in a simulation during 1 s. Curve 2a in Figure 35 depicts the evolution of the fundamental A. The new origin of time for the curves 1a and 2a is considered to be at a zero-crossing moment (from negative to positive values) of the fundamental A. Before the addition of the sinusoidal components, all the phases at origin of time for components A, A_1_ ÷ A_6_ from Table 2 are recalculated relative to this new origin of time. Curve 3a describes in the same way the evolution of the variable part of IAP_1_ (IP) generated by the same flat belt 1 after 100 s of supplementary running in idle of the gearbox. It is obvious that there is quite a good similarity between the curves 1a and 3a, with some relatively minor changes, probably due to the rising temperature of the belt and pulleys.

By comparison with the evolution depicted by curve 1a (from Figure 35), curve 1b in Figure 36 presents the description of flat belt 1’s influence on the variable part of IAP_1_ (or IP as well) if, in the gearing diagram (Figure 2), the electromagnetic clutch EMC3 is disengaged. Now, the rotation of shafts II and III and the output spindle are stopped, the AC motor drives only flat belt 1 and shaft I, and the average value of the IAP1 decreases from 3067 W (related to Figure 35) to 1021 W (related to Figure 36). Apart from changing the aspect of evolution (1b versus 1a), the amplitude of the fundamental (37.99 W on curve 2a) strongly decreases (only 5.5 W on curve 2b).

Here *f_A_* = 5.390 Hz and *f_Aa_* = 5.388 Hz. In Figure 36, there is a small progressively decreasing time delay between the curves 1a and 1b because the value *f_Aa_* is slightly different (5.336 Hz for curve 1a compared to 5.388 Hz for curve 1b). The decrease in the mechanical loading of the AC induction motor (due to the fact that the electromagnetic clutch EMC3 is disengaged) produces a slight increase in rotation speed, and the increase in the frequency *f_A_* (and *f_Aa_* as well) is due to the decreasing motor slip.

We should mention that (with the electromagnetic clutch EMC3 disengaged) the dynamic model of the AC motor’s rotor, which is rotated by a rotary magnetic field, now has a torsional vibration mode of 10.785 Hz in frequency (with an 0.308 damping ratio). With *f_A_*_1_ = 10.77 Hz, there is a resonant amplification for the first harmonic A_1_ (having a 26.95 W amplitude). Because of structural damping, this resonant amplification strongly decreases when the entire gearing diagram is used (in Figure 35, on curve 1a, and in Table 2, the same first harmonic A_1_ has its amplitude diminished at 15.48 W).

Similarly, by the mathematical addition of sine components B, B_1_, B_2_ and B_3_ (having their description revealed by curve fitting, but with the frequency of harmonics *f_Bai_* = (*I* + 1)*·f_Ba_* and *f_Ba_* = 9.4423 Hz calculated in the same way as in Equation (4)), it is possible to find the evolution of the variable part of IAP_1_ (or IP as well) generated by flat belt 2, as Figure 37 indicates (with the fundamental in red).

The analysis by curve fitting allows a comparative study of the variation produced by flat belt 2 on IAP_1_ (curve A in Figure 38, as a short sequence from Figure 37) and on IAS (curve B in Figure 38) during the same experiment. There is a difference of shapes and a shift of phase between the curves because IAP_1_ is measured on the AC motor (at the gearbox input), and IAS is measured at the output spindle (at the gearbox output). The gearbox introduces different attenuations (and different phase shifts) for the influence of flat belt 2 (on IAP_1_ and IAS) towards the AC motor and towards the output spindle.

The evolutions from Figure 35, Figure 36 and Figure 37 are useful in the condition monitoring of the flat belt transmissions (available for many other types of belts). There is a simple possible approach: if the peak-to-peak value of the IAP_1_ (IP) variable part generated by a belt (or the amplitude of the fundamental or some harmonics) exceeds an imposed limit, then the end of its proper working life has been reached. This is clearly proved by a new experiment. The headstock gearbox runs in idle in the configuration already depicted in Figure 2 and used in the previous experiments. This time, a damaged flat belt 1 (more than 40 years old) was used.

Figure 39 presents a partial view of this flat belt, finally broken during the experiments.

Firstly, the belt was partially broken only in area A, due to a mistake made when it was reinstalled on the pulleys. A severe decrease in the belt’s stiffness is produced in that area. For the time being, the belt is not yet broken in area B. The description of the variable part of IAP_1_ (IP) generated by this damaged flat belt was determined in a similar way to those depicted in Figure 35 (for curve 1a) and described with curve 1c in Figure 40.

Here, curve 2c describes the evolution of the fundamental. A comparison between Figure 35 and Figure 40 reveals a big increase in the peak-to-peak evolution of the variable part of IAP_1_ (IP) generated by the damaged belt compared with a regular one, and a big increasing in the amplitude of the fundamental (233.704 W here, in contrast with 37.99 W in Figure 35).

A better comparison between curve 1a (from Figure 35, which describes the behaviour of a regular flat belt) and curve 1c (from Figure 40, which describes the behaviour of a damaged flat belt) is allowed by Figure 41. The same magnitude was used for both curves.

It is obvious that damage to a flat belt can be detected a long time before a catastrophic failure. We should mention that this flat belt was completely broken much later in area B (Figure 39) during an experiment with an abnormally severe transient regime of gearbox acceleration.

We should highlight an important matter: the comparison between curves 1a and 3a in Figure 35, or between curves 1a and 1b in Figure 36, or between curves 1a and 1c in Figure 41 (each curve being the result of a different experiment) was possible (related to the phase relationships between the curves) firstly because each one has its origin on time strictly (recalculated) at a zero-crossing moment of its fundamental (curve 2a, 2b and 2c respectively) and secondly because the fundamentals have, with a good approximation, the same frequency.

This type of condition monitoring can be also applied for other types of drive belts (e.g., v-belts, in some preliminary experiments).

In the same way, it is possible to produce good descriptions of the behaviour of some other parts of the gearbox (especially the shafts) and to detect abnormal operating situations using mainly the variable parts of IAP_1_ (IP).

As a first example, the ordinates of some samples of the AE_IP_ evolution from Figure 22 were recalculated by curve fitting (as AE_PCF_ samples). Each AE_PCF_ sample (described as a red rectangle in Figure 42, a completion of Figure 22) describes an average of the amplitude of peak E during 5 s of the evolution of IAP_1_, determined by curve fitting (the spectra of IP and IAP_1_ being identical). The fact that there is no perfect fit between the AE_IAP1_ samples and the average AE_IP_ indicates once more that the IP_E_ (or IAP_1E_ as the variable part of IAP_1_ related to peak E) has a noisy evolution, partially because the frequency *f_E_* of the peak E on IAP_1_ (IP) is not strictly constant (another issue involved in condition monitoring). The average values of this frequency for each AE_IAP1_ sample (also a result of curve fitting process) are written in Figure 42.

As a second example, the ordinates of some samples of the AE_Vs_ evolution (curve 2 from Figure 20, depicted in bright blue in Figure 43) were recalculated by curve fitting from the VS_S_ signal (as AE_VsCF_ samples).

Each of the 40 timewise equidistant AE_VsCF_ samples (depicted as pink rectangles in Figure 43) describes an average of the local amplitude of peak E during 1 s of evolution of VS_S_, determined by curve fitting. This time, there is an almost perfect fit between the AE_Vs_ evolution and the AE_VsCF_ samples (due to the fact that the component E is dominant in vibration). This means, first, that both methods for finding the amplitude of the peak E on VS_S_ (the FFT spectra on short movable sequences SMSVs and the curve fit) work quite well. Secondly, this means that, in contrast to the AE_IP_ (or AE_PCF_) evolution, the AE_Vs_ (and AE_VsCF_ as well) evolution does not have a noisy evolution. In the amount of power used to rotate shaft III and the spindle (*p_E_* = *p_vdp_* + *p_ep_*) the AE_IP_ (noisy) evolution is strictly related to the power *p_E_*, while the AE_Vs_ (noiseless) evolution is strictly related to the power *p_ep_*. It follows that the power *p_vdp_* also has a noisy evolution.

The determination of the mathematical description of the main signals components (sum of sinusoids model) by curve fitting can be successfully applied for condition monitoring purposes based on the gearbox vibration [49,50] (partially proved in Figure 43) or instantaneous angular speed evolutions (partially proved in Figure 38).

## 5. Discussion

This work reveals the behaviour of a three-phase AC asynchronous induction motor as a sensor of mechanical power useful for the condition monitoring of driven machines with the different mechanical parts inside having periodical rotary motions (with a constant rotation speed). An obvious argument justifies this approach: the mechanical power delivered by motor at the rotor is firmly mirrored in the input (active) electrical power absorbed from the supplying electrical network. Each rotary part inside the driven machine needs mechanical power to supply the motion in idle. When this motion has a constant speed, the variable part of this mechanical power is frequently periodical, described as a sum of sinusoidal components with a fundamental and some harmonics. This mechanical power is mirrored in the electrical power, and the mathematical descriptions of its components are detectable (by some computer-assisted procedures) and useful for the condition monitoring of that rotary part.

A particular rotary machine was considered here: a headstock lathe gearbox with shafts, belts, gears, bearings, etc., running idle with constant rotational speed. A simple computer-assisted experimental setup is used for data acquisition and signal processing. The block diagram from Figure 44 briefly depicts the main steps, flows, procedures and results in signal processing focused on the gearbox condition monitoring performed in this work.

The evolutions in time of the instantaneous voltage (IV) and the absorbed instantaneous current (IC) on a single phase acquired as AC signals (using a VT and a CT transformer) from the electrical input of the AC electrical motor (supplied directly with a three-phase four-wire system) are used for computer-assisted signal processing in order to achieve the descriptions of some constituents of electrical power: mainly, the instantaneous electrical power, instantaneous active electrical power and instantaneous power factor. This simple setup (possible to be greatly improved in the future by replacing the CT transformer with a current probe for an oscilloscope) has a big advantage: it can be very easily used for the electrical power monitoring of any other AC induction motor.

The AC signal delivered by a vibration sensor VS (as VS_S_) placed on the gearbox and the AC signal delivered by an instantaneous angular speed sensor IASS (as IASS_S_) placed on the output spindle are also used as alternative, comparative methods for condition monitoring. These four AC signals are acquired simultaneously in numerical format (using a numerical oscilloscope) with a high sampling rate and a high number of samples. This allows a high value for the Nyquist limit and a high resolution in frequency when the signals are described and analysed in the frequency domain for diagnostic purposes by the fast Fourier transform (FFT), a privileged topic in our work (mainly because the gearbox parts rotate with constant speed).

Using the IC and IV signals described in numerical format, a definition of the instantaneous electrical power (IP), two definitions of the instantaneous active electrical power (IAP_A1_ with low sampling frequency and IAP_A2_ with high sampling frequency) and a definition of the instantaneous power factor (IPF) were proposed and used in order to achieve the description of the evolution in the time and frequency domains (by FFT), using Matlab, for the main constituents of the electrical power absorbed by the AC induction motor.

First, a study on this gearbox establishes that, in a relative low area of frequencies, the spectra of IP, IAP_A1_ and IAP_A2_ are practically the same; however, the IP spectrum is preferred, mainly because IAP_A1_ has a low Nyquist limit at 50 Hz, and IAP_A2_ introduces some attenuations of the peak amplitudes at high frequencies (IAP_A2_ being the result of a low pass filtering of IP using a moving average filter). However, the IP spectrum has a relative important disadvantage: it contains high amplitude peaks, which describe mainly the harmonics of the fundamental on 100 Hz (electrically generated).

The resources of condition monitoring on this rotary machine running idle with a constant speed were particularly revealed in this work by the contents of FFT spectra in low area of frequencies (0 ÷ 40 Hz). The FFT spectrum of the IP (here with a 0.01 Hz frequency resolution) proves, firstly, that the IP can be approximated quite well as being a deterministic signal. Secondly, it was revealed that the IP spectral peaks describe the behaviour of some gearbox parts (belts, shafts and output spindle in a first approach). More precisely, they describe the average amplitudes of the components in a sum of sinusoids model for the variable part of the instantaneous electrical (or active) power absorbed by the AC motor and consumed to rotate the gearbox parts in idle. The amplitude of these peaks in the IP spectrum is an important feature involved in condition monitoring.

This work also proves that the FFT spectra of some other electrical parameters involved in the description of the electrical power flowing through the AC motor fully confirm the availability of the IP spectrum (all the peaks are described similarly on all spectra, at least related to frequencies). First is the FFT spectrum of IPF. Second (as a new proposal) is the FFT spectrum of full-wave rectified instantaneous current (RIC), a substitute for the FFT spectrum of IC (not relevant in our work), currently claimed to be used in the literature. Third (also as a new proposal) is the FFT spectrum of full-wave rectified instantaneous voltage (RIV), similarly a substitute for the FFT spectrum of IV. This last approach is not very relevant; the availability of RIV depends a lot on the voltage drop on the wires used to supply the AC motor, or on their electrical resistance. However, we consider that the FFT spectrum of IP is the best approach available to describe the condition of a driven rotary machine (as being more closely related to the variation of mechanical loading).

The availability of the FFT spectrum of IP was also fully confirmed by two supplementary spectra generated by the signals of other sensors placed on the gearbox: the spectrum of VS_S_ and the spectrum of IAS during the same experiment. All the peaks are described similarly on all spectra, at least related to frequencies. For example, a spectrum generated by the pointwise multiplication (product) of the IP and VS_S_ spectra fully confirms the similarities. The similarities between the IP and IAS spectra prove that there is a deterministic relationship between the mechanical loading and the angular speed of the motor: the speed decreases when the torque increases, and vice versa, due to the motor slip. The similarity between the IP and VS_S_ spectra proves that many components of the forced vibration motion of the gearbox are powered by mechanical power delivered by the AC induction motor, especially those provided by mass imbalances of the rotating shafts.

This last similarity was supplementary investigated. It was proved that a low frequency (or high period: 96 s) beating phenomenon of vibration superposition in the gearbox (in the VS_S_ signal) generated by the rotation with a mass imbalance and with almost the same speed of the output spindle and a shaft is powered by a beating phenomenon in the power from the periodical components inside the IP. A computer-aided method was proposed and developed in order to find the evolutions of the amplitude of the beating vibration and the amplitude of the beating power. First, the frequency of the beating phenomenon (the same in VS_S_ and IP) was detected (17.37 Hz). The amplitude evolution of the peak at this frequency in the FFT spectra of a short movable sequence (with overlap) of the VS_S_ and IP evolutions (having the same sampling frequency *F_s_*) is the same as the amplitude of the beating phenomenon. The number of samples in the short movable sequences (*N_ss_*) is calculated in such a way that the *k*th sample in the FFT spectrum of the short movable sequence (having the frequency *k·F_s_*/*N_ss_*) has the closest possible value to the frequency involved in the beating phenomenon. Each movable sequence produces a value of amplitude.

The shift of phase (time delay) between the beating phenomenon mirrored in instantaneous power and in vibration was also investigated with some supplementary experiments which modify this shift of phase by changing the vibration conditions of the gearbox or by introducing a supplementary periodical mechanical loading at the spindle (generated by a temporary interrupted cutting process). This work proves that this beating vibration phenomenon is well mirrored in the IAS and RIC evolutions, too. As an interesting achievement, this work proves that some important properties of the AC induction motor used as a sensor (related to a variable mechanical loading) are more relevant by comparison with the properties of a vibration sensor. However, it was indirectly proved that, because all periodical forced vibrations should be mechanically powered, there are two ways to describe these vibrations: using a vibration sensor or using the mechanical power measurement at the excitation source.

The FFT spectra of IP or RIC and eventually VS_S_ in high frequency areas are available for the condition monitoring of other parts of the gearbox (e.g., gears, bearings). The IP or RIC evolutions in the time domain are also available for many other techniques of signal processing proposed in the literature for almost periodical signals.

This work also introduces a proposal for condition monitoring: the determination of a completely mathematical description of the main sinusoidal components (amplitude, frequency and phase at origin of time) inside the variable part of IAP_1_ (the instantaneous active power absorbed by the motor, three times bigger than IAP_A1_) by curve fitting (as a signal components identification procedure) based on the sum of sinusoids model, using two methods (both in Matlab) with similar results: a method proposed by us (slower) and the Curve Fitting Tool application (faster). The availability of these methods was experimentally proved firstly by comparison of the FFT spectra for a short experimental IAP_1_ sequence before and after the main sinusoidal components were removed and secondly by comparison in the time domain of this experimental sequence with the theoretical one built as a sum of the determined sinusoidal components.

This method allows an important approach to condition monitoring: it is possible to find the evolution in the time domain of the variable part of IAP_1_ (or IP) introduced by some mechanical parts inside the gearbox (as a sum of the detectable sinusoids with a fundamental and some harmonics) as a health condition characterization. In order to facilitate the comparison, related to the behaviour of a mechanical part in different circumstances, the origin of time of this variable part of IAP_1_ is placed in a zero-crossing moment of the fundamental, e.g., from negative to positive values.

As an example, the evolution of the periodical part of the power introduced by a regular flat belt placed at the gearbox input was described in different circumstances. Firstly, a quite good autocorrelation between two different sequences of these periodical parts of the variable power (5 s in duration each with a 100 s delay between) was proved. Secondly, the significant difference between the periodical parts of the power introduced by the same belt for two different gearing states of the gearbox was highlighted experimentally. Thirdly, a high difference was emphasised between the amplitude of the periodical parts of the power introduced by a seriously damaged belt compared with a regular one, using the same gearing state of the gearbox running idle.

The curve fitting method is appropriate also for describing the periodical components (as a sum of sinusoids) inside the vibration or instantaneous angular speed signals.

An essential observation can be made: if the average IAS in this stationary idling regime of the gearbox is not constant, then condition monitoring using both methods (based on FFT or curve fitting) is not conceivable, mainly because the variable part of IP, IAP_1_, IAS, VS_S_, RIC and IPF can no longer be described as a sum of sinusoids with constant frequencies. Two factors imply the variation of the IAS. Firstly, typically for AC asynchronous induction motors, the variation of the IAS is related to the slip of the motor due to the variation of mechanical power, indirectly measured here through the active electrical power. Secondly, this variation can be generated by the variation of the frequency of the electrical power supply system, because the synchronous speed of the motor depends on this frequency.

The dependence of the IAS on mechanical power is indirectly experimentally revealed as follows. During the experiments with the results depicted in Figure 22, two evolutions were also collected simultaneously: the low pass filtered evolution of the IAS at the output spindle (theoretically strictly related to the AC induction motor IAS due to the gearing diagram, Figure 2) and the low pass filtered AEP_2_. This AEP_2_ is the total active electrical power absorbed by motor, three times AEP_A2_, and theoretically proportional with the mechanical power delivered by motor. The dependence of this IAS on AEP_2_ is depicted in Figure 45. Here, the low pass filtered AEP_2_ decreases in time (and mechanical power, too) mainly because of gearbox heating. As a consequence, as the low pass filtered IAS increases, the motor slip decreases.

Contrary to any expectation, this dependence is not describable by a simple portion of curve (eventually reducible at a line), mainly because the IAS is not measured at the AC induction motor output rotor. In Figure 45, there are many irregular loops, probably because of the irregular slipping of the belts and clutch. However, an approximation of this evolution by a linear regression line reveals a very small negative slope (rate of IAS) of 0.8953 rad/(s·KW) or of 0.1424 Hz/KW as rotational frequency. This small value of the IAS slope (less than 0.2% in Figure 45) confirms that, with a good approximation, the IAS evolution (and the rotational frequency, as well) can be considered constant.

Figure 46 presents the evolution of the frequency of IV (filtered and unfiltered) during the same experiment. This IV frequency was accurately measured based on the same procedures used before for the IAS measurement [46] (the IASS_S_ signal was replaced by the VT_S_ signal). The unfiltered evolution of the IV frequency has a small peak-to-peak maximum variation of only 0.1262 Hz, and only 0.0583 Hz on low pass filtered evolution. Because the AC asynchronous induction motor has two magnetic poles, the peak-to-peak variation of its synchronous rotational frequency is 0.1262/2 = 0.0631 Hz (for IV unfiltered frequency evolution) and 0.0583/2 = 0.0291 Hz (for filtered evolution). In the worst possible scenario (the synchronous rotation frequency follows the IV unfiltered frequency evolution), the peak-to-peak variation of the rotation frequency is less than 0.26%. The influence of the IV frequency can be also neglected.

We should mention that the variation of the supplying frequency highlighted in Figure 46 exerts a (small) bad influence on the evolution of the filtered IAS, as already depicted in Figure 45. Nevertheless, if necessary, at least the curve fitting procedure can eliminate these small shortcomings by using short signal sequences for analysis. During these short sequences, the sum of sinusoids model for the signals IAP_1_, IAS, VS_S_, RIC and IPF can be considered viable.

This relatively small variation of the motor speed in stationary idle regimes due to the slip phenomenon is a big advantage, as it allows the condition monitoring of the driven machine as it is equipped, frequently, with an AC asynchronous induction drive motor, used as sensor. Otherwise, the use of a synchronous drive motor should be mandatory during condition monitoring procedures.

## 6. Conclusions and Future Work

Some important achievements in the condition monitoring of rotary machines driven by a three-phase AC asynchronous induction motor, running in idle with constant rotary speed, are revealed in this paper:The AC induction motor used to drive a rotary machine works properly as a sensor of the variable (periodical) mechanical power phenomena involved in condition monitoring, through the absorbed electrical power.The variable part of some different constituents of the electrical power is available for condition monitoring: e.g., instantaneous and active electrical power, the power factor.The resources for condition monitoring using the full-wave rectified instantaneous current (RIC) and full-wave rectified instantaneous voltage (RIV) were revealed.A very important feature of the electrical power constituents was highlighted: they have a preponderantly deterministic evolution, mathematically described as a sum of sinusoids with a low level of noise.Two known signal processing methods were defined in order to find out the mathematical description of each significant term of this sum (with the values of amplitude, frequency and phase at origin of time), which are well related to the normal or abnormal behaviour of a rotary mechanical part through its rotation frequency.The highlighting of the possibilities offered by these condition monitoring methods was accomplished through an experimental study of a beating phenomenon in instantaneous power and a flat belt behaviour.Any type of gearbox failure which causes a variable (periodic) consumption of mechanical power during the idle running regime can be detected using condition monitoring based on an AC induction motor used as loading sensor, through the analysis of the absorbed electrical power and its components by FFT or curve fitting, if this variable consumption mirrored in the electrical power is bigger than the measurement noise. For example, in this paper, the behaviour of a damaged flat belt (in comparison with a regular one) was clearly revealed. Also, the behaviour of some shafts and the output spindle was described. However, we should mention that the gearbox moment of inertia works as a low pass mechanical filter. This makes the condition monitoring of relatively high frequency mechanical phenomena generated by rotary parts difficult.The experimental approaches and their achievements were fully confirmed and validated by the evolutions of the vibration of the gearbox and the instantaneous angular speed of the output spindle.

The first challenge for a future work will be the optimisation of the IP (IAP_1_) signal analysis by computer aided curve fitting as a programmed procedure in Matlab, in order to drastically reduce its duration.

Future work will also focus on the condition monitoring of rotary machines and mechanical parts based on the exploitation of the IP (or RIC) resources at higher frequencies, useful for the condition monitoring of gears and bearings. The same resources will be used to detect and to describe the excited torsional vibration modes of shafts, because these vibrations (involved in gearbox dynamics) are also mechanically powered by the AC induction motor.

A computer-assisted method (by curve fitting) will be developed in order to automatically remove the electrically generated fundamentals (at the 50 and 100 Hz frequencies) and their harmonics from the IP (or RIC) evolution in the time domain. Secondly, an automatic procedure will be developed in order to detect the description of the variable part of the IP introduced by each mechanical part using a curve fitting procedure (as was previously performed for flat belt 1). Some other new possible features of the electrical power will be exploited: the description with a high sampling rate of the IPF evolution (bigger than 100 s^−1^), the use of instantaneous reactive power (IRP) in condition monitoring, based on a possible relationship of IRP with the periodically excited vibrations inside the driven rotary machines, as the literature suggests [51], etc.

A privileged topic in future research will be the exploitation as sensor of mechanical loading for an AC induction motor electrically supplied with an AC to AC converter.

## Figures and Tables

**Figure 1 sensors-23-00488-f001:**
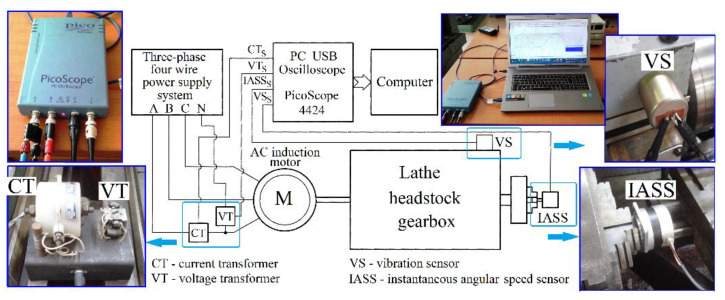
A description of the experimental setup.

**Figure 2 sensors-23-00488-f002:**
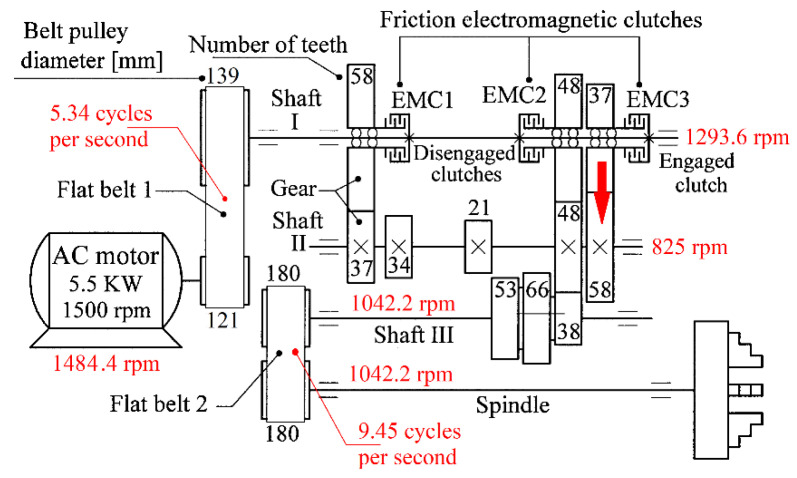
The gearing diagram of the lathe headstock gearbox used in experimental setup.

**Figure 3 sensors-23-00488-f003:**
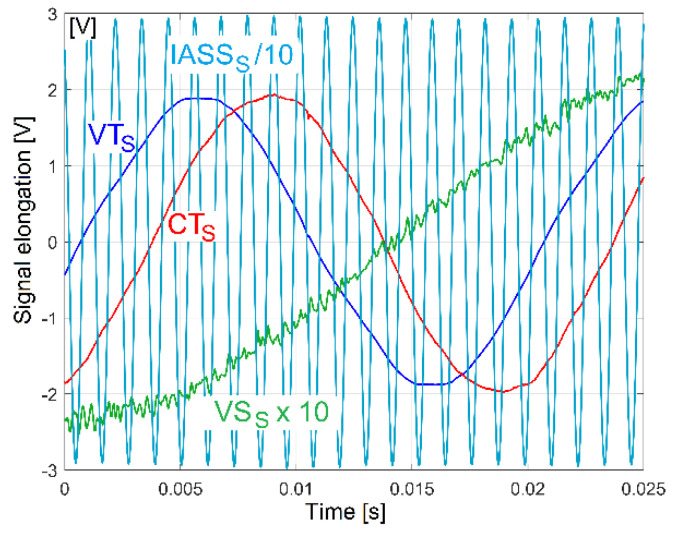
A short sequence with the evolution of VT_S_, CT_S_ signals (generated by VT and CT transformers) and VS_S_, IASS_S_ signals (generated by VS and IASS sensors).

**Figure 4 sensors-23-00488-f004:**
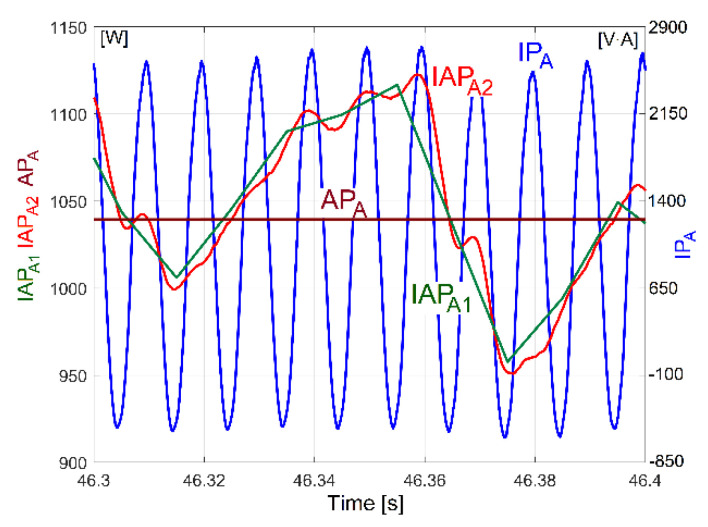
A short sequence (0.1 s) with the evolutions of IP_A_, IAP_A1_, IAP_A2_ and AP_A_ with lathe headstock gearbox running idle.

**Figure 5 sensors-23-00488-f005:**
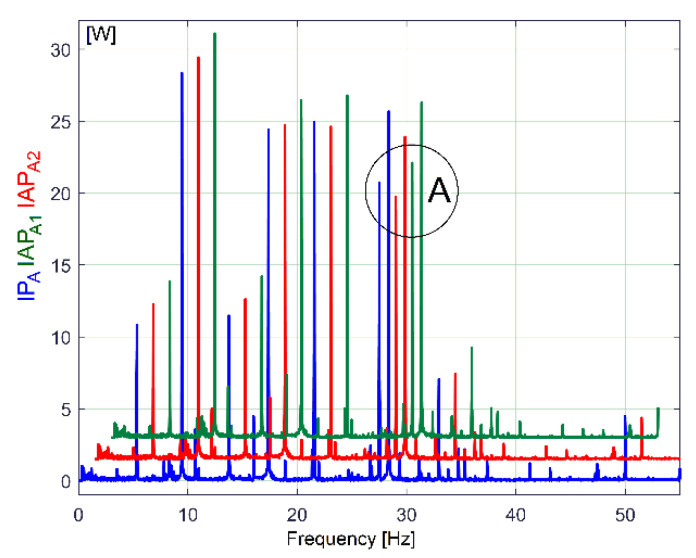
A short sequence (0 ÷ 55 Hz) with FFT spectra of IP_A_, IAP_A1_ and IAP_A2_ with headstock gearbox running idle.

**Figure 6 sensors-23-00488-f006:**
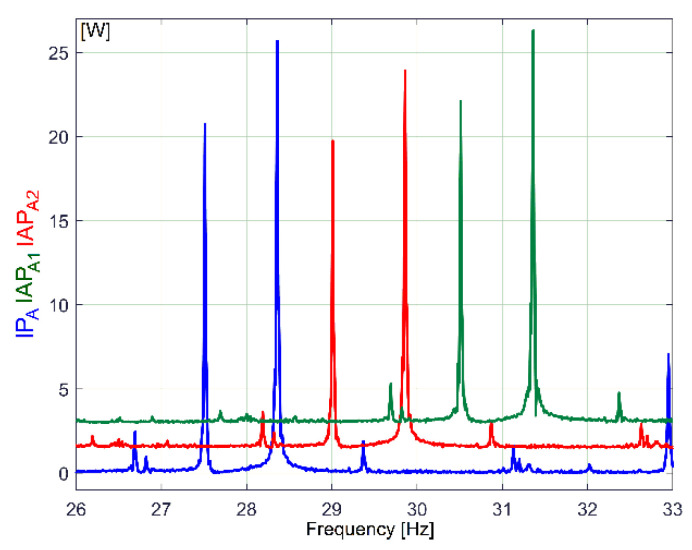
A zoomed-in detail of Figure 5 (in area A).

**Figure 7 sensors-23-00488-f007:**
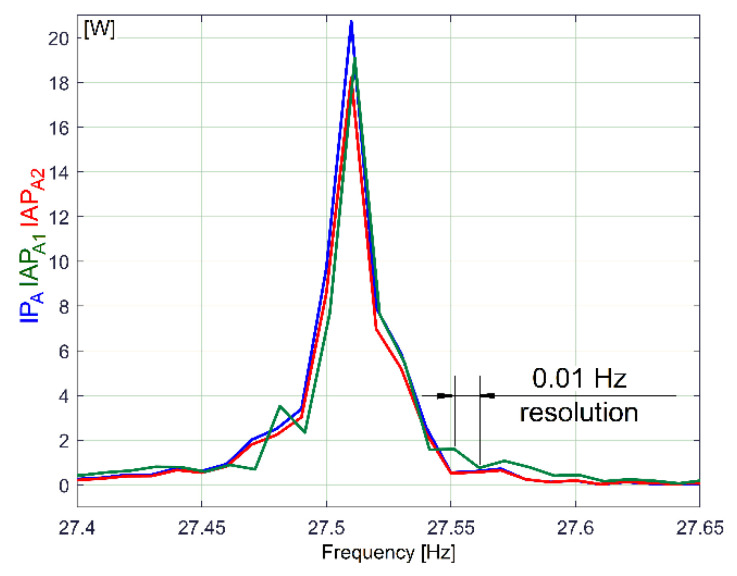
The IP_A_, IAP_A1_, IAP_A2_ evolutions (not shifted) at first significant peak in Figure 6.

**Figure 8 sensors-23-00488-f008:**
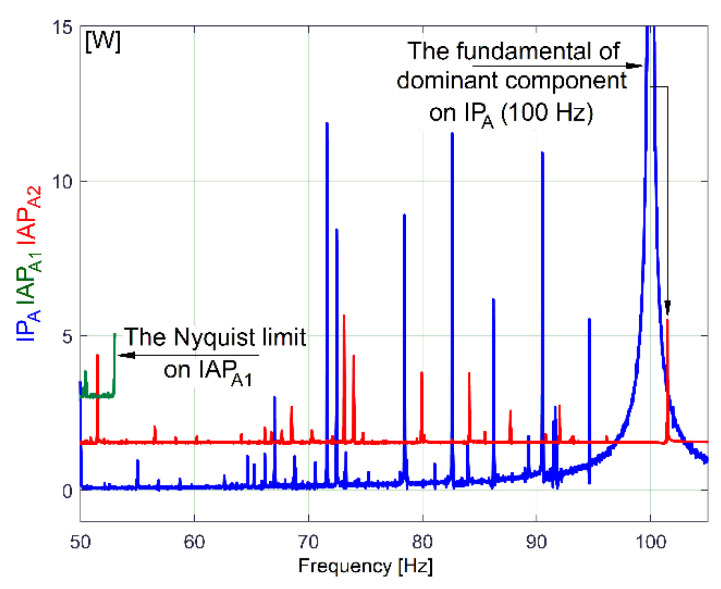
A short sequence (50 ÷ 105 Hz) with FFT spectra of IP_A_, IAP_A1_ and IAP_A2_ with gearbox running idle.

**Figure 9 sensors-23-00488-f009:**
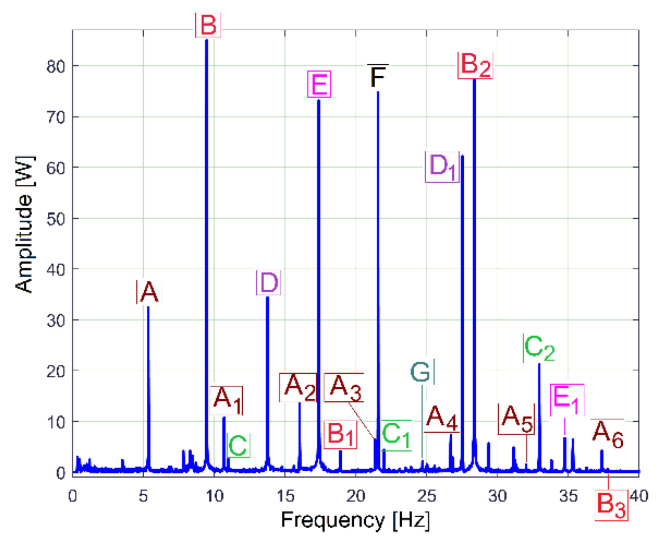
A short sequence of FFT spectrum of IP and the identification of the main components (fundamentals and harmonics) generated by some gearbox parts.

**Figure 10 sensors-23-00488-f010:**
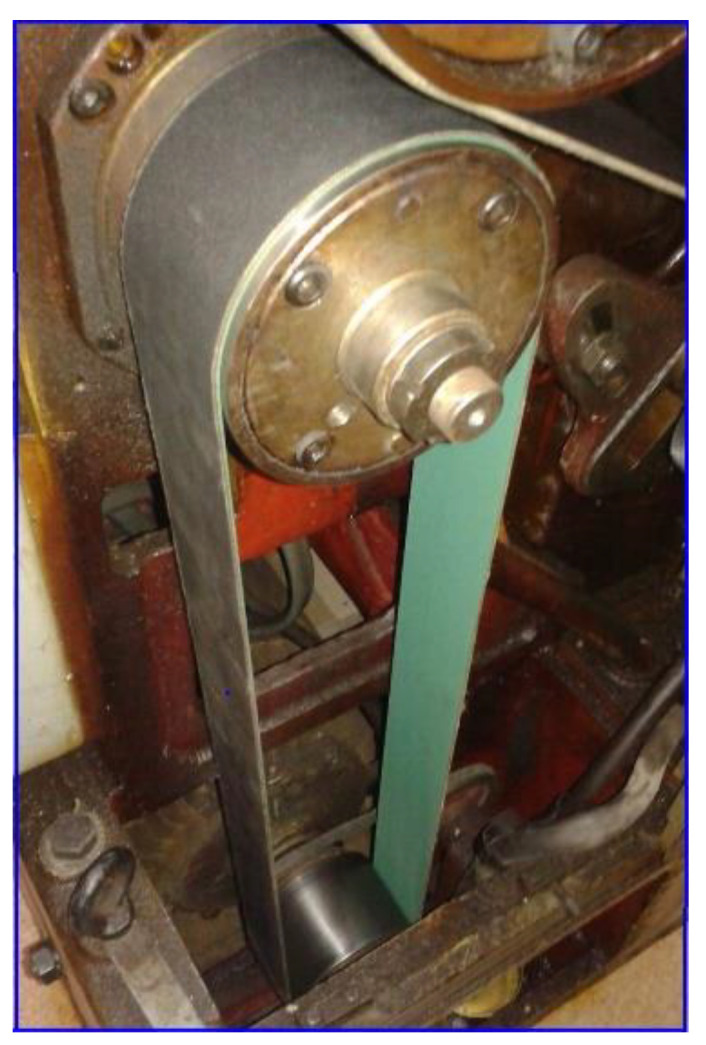
A view of flat belt 1 placed on its pulleys.

**Figure 11 sensors-23-00488-f011:**
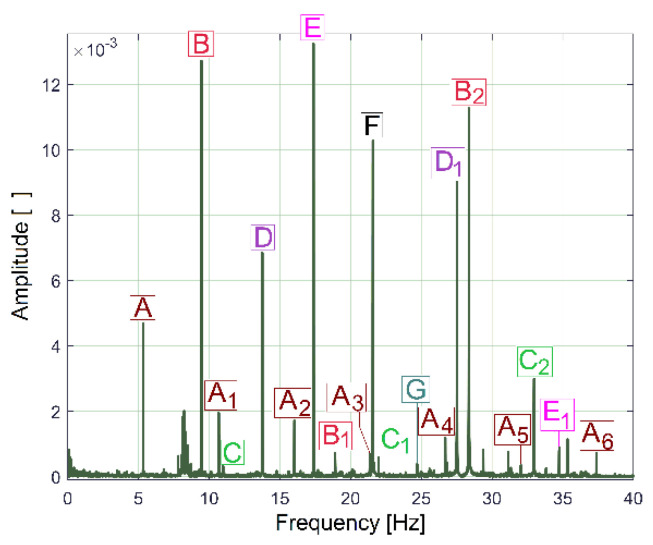
The behaviour of gearbox parts described in the FFT spectrum of IPF (for comparison with Figure 9).

**Figure 12 sensors-23-00488-f012:**
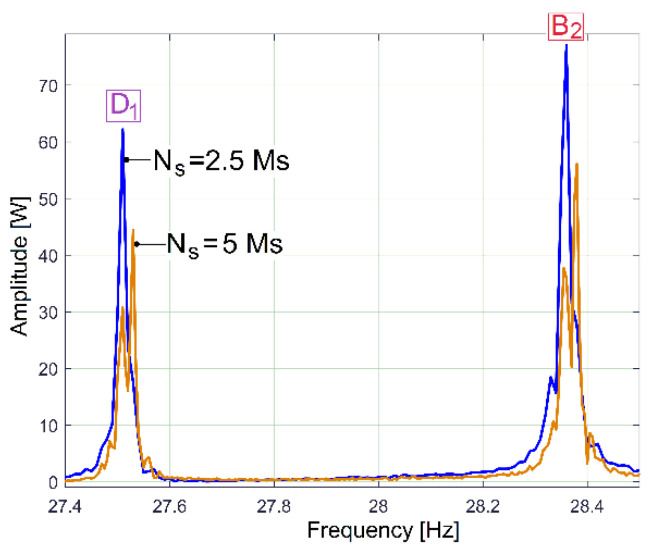
The influence of number of samples *N_s_* on the quality of FFT spectra for IP.

**Figure 13 sensors-23-00488-f013:**
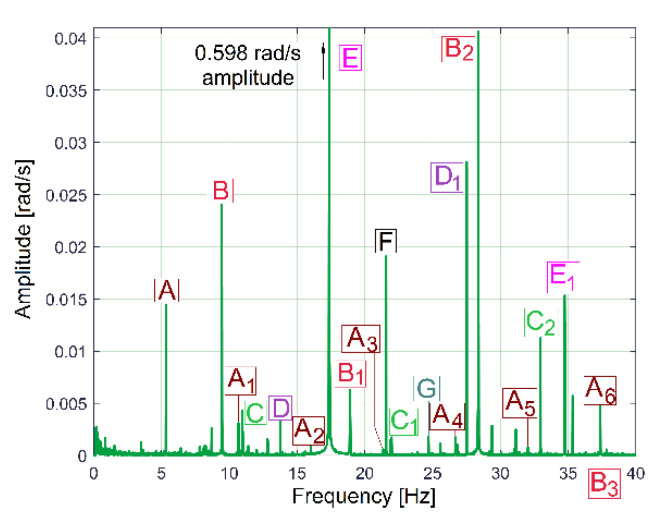
The behaviour of gearbox parts described in the FFT spectrum of IAS (for comparison with Figure 9).

**Figure 14 sensors-23-00488-f014:**
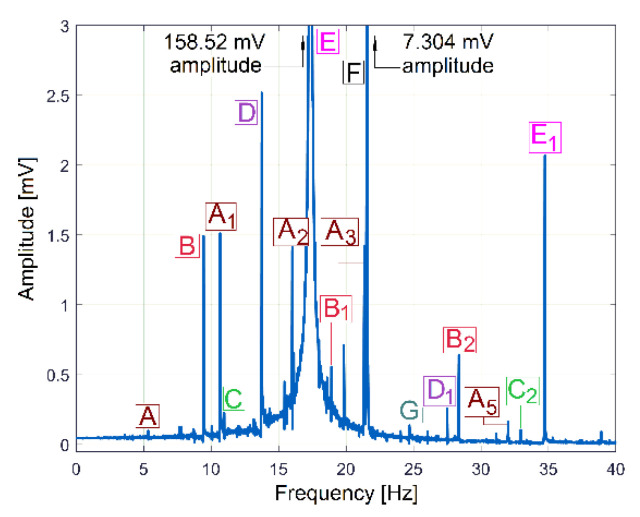
The behaviour of gearbox parts described in the FFT spectrum of vibration signal VS_S_ (for comparison with Figure 9).

**Figure 15 sensors-23-00488-f015:**
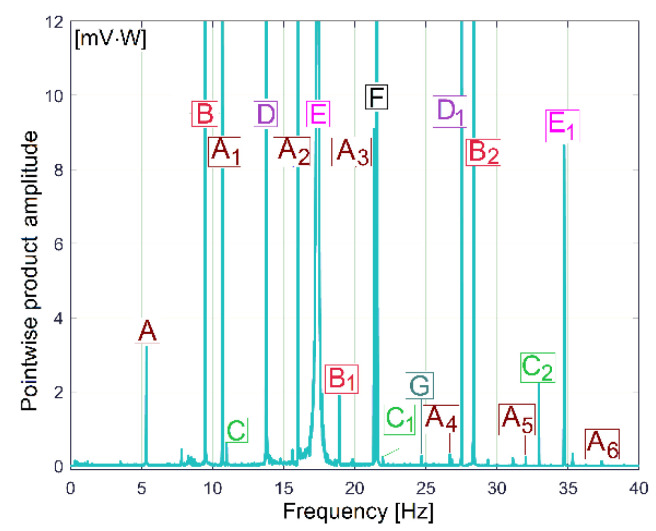
A partial view on the pointwise multiplication (product) of IP and VS_S_ spectra.

**Figure 16 sensors-23-00488-f016:**
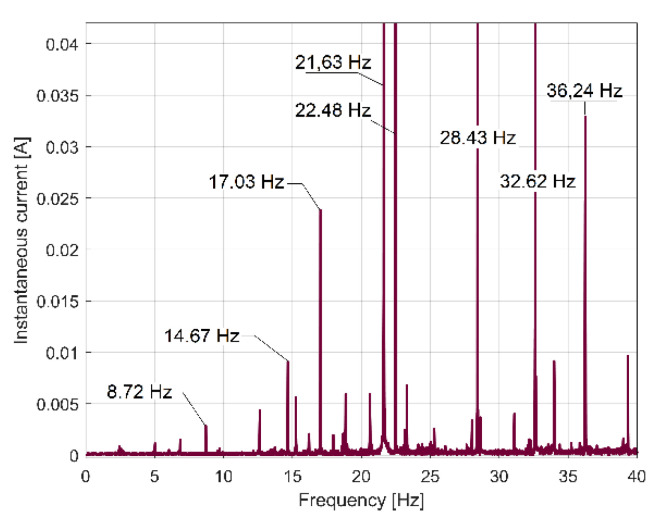
A short sequence of FFT spectrum of IC.

**Figure 17 sensors-23-00488-f017:**
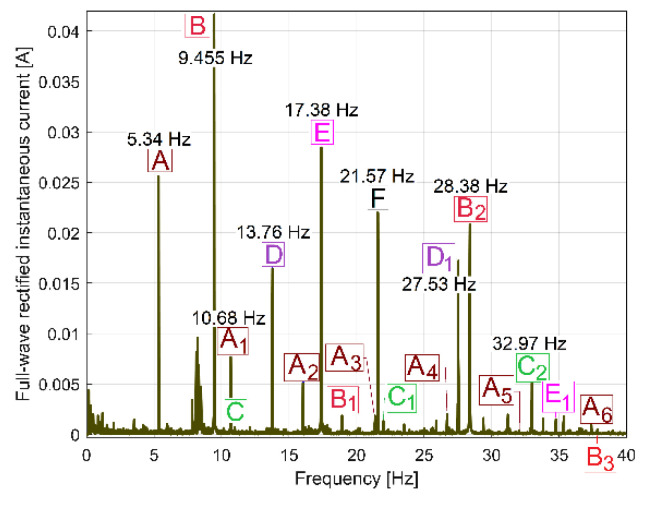
The behaviour of gearbox parts described in the FFT spectrum of RIC (for comparison with Figure 9).

**Figure 18 sensors-23-00488-f018:**
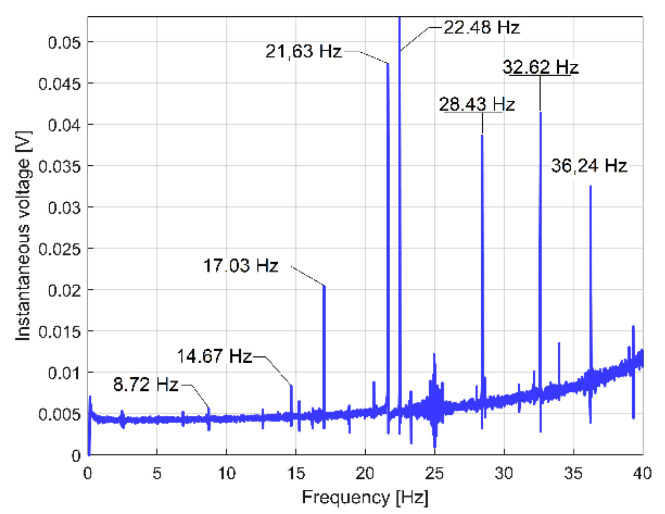
A short sequence of FFT spectrum of IV.

**Figure 19 sensors-23-00488-f019:**
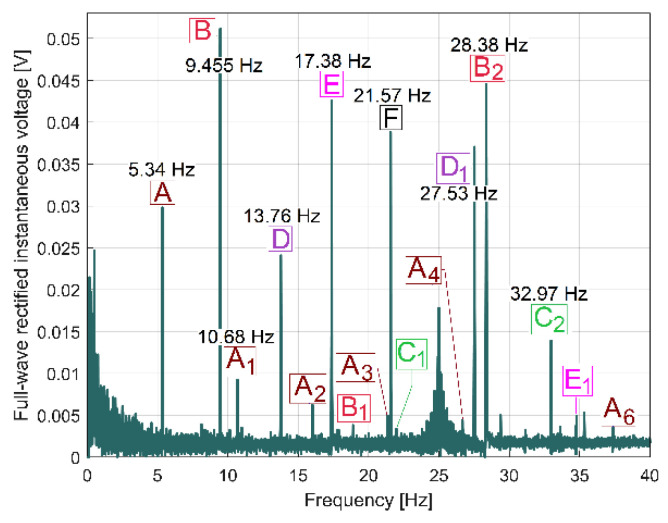
The behaviour of gearbox parts described in the FFT spectrum of RIV (for comparison with Figure 9 and Figure 17).

**Figure 20 sensors-23-00488-f020:**
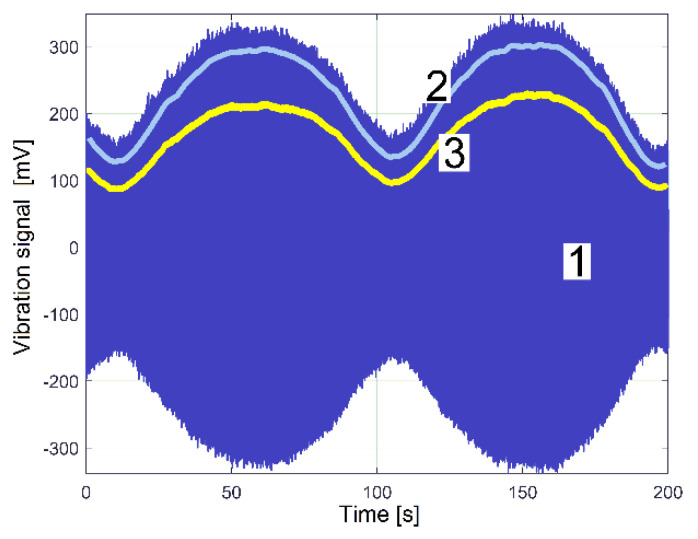
Beating phenomena in vibration signal VS_S_ (1), with correct (curve 2) and wrong (curve 3) description of the evolution for the amplitude AEVs of the dominant component (E).

**Figure 21 sensors-23-00488-f021:**
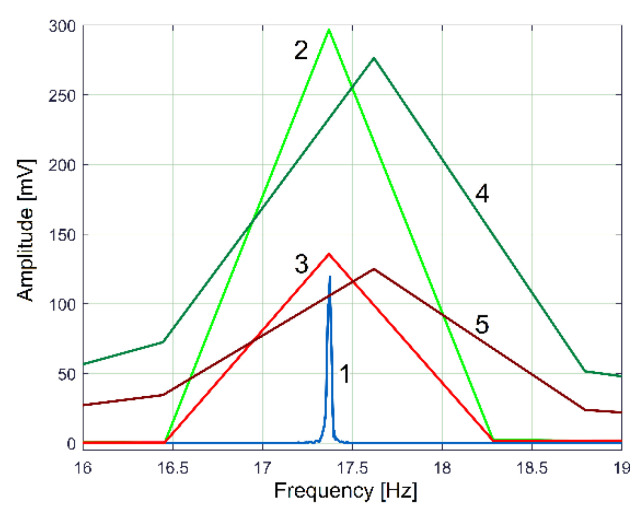
A comparison of FFT spectra features for the peak E: on entire VS_S_ (curve 1), using appropriate SMSVs size (curves 2, 3) and a wrong SMSVs size (curves 4, 5).

**Figure 22 sensors-23-00488-f022:**
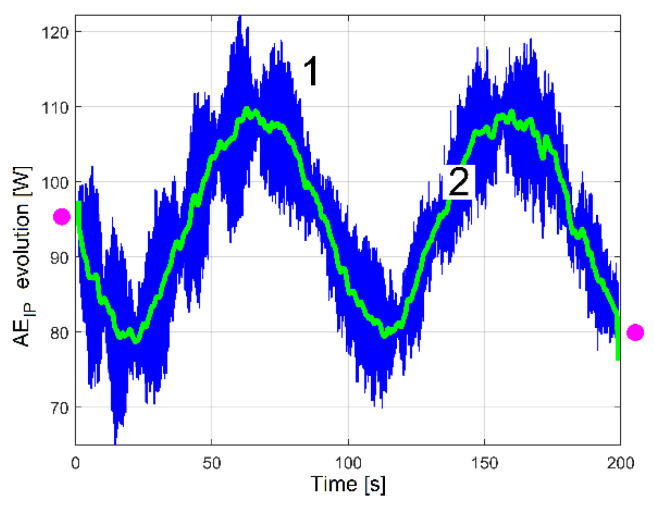
The AE_IP_ evolution during the experiment depicted in Figure 20, without filtering (curve 1) and low pass filtered (curve 2, as moving averaged AE_IP_). The pink dots marks some distorted data (due to filtering edge effects).

**Figure 23 sensors-23-00488-f023:**
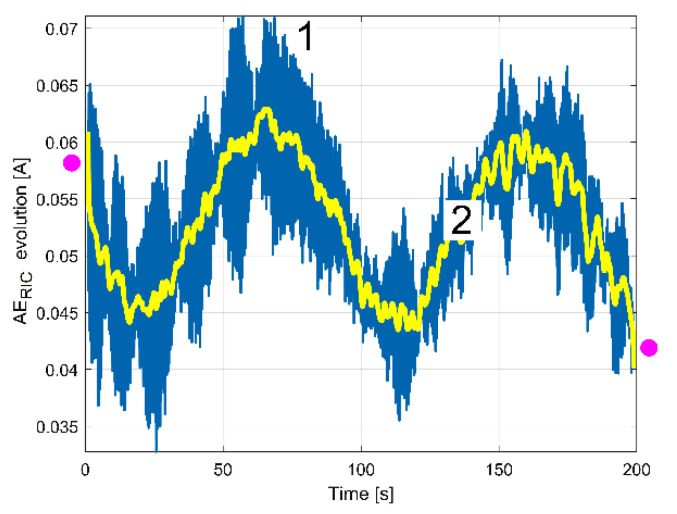
The AE_RIC_ evolution during the experiment depicted in Figure 20, without filtering (curve 1) and low pass filtered (curve 2, as moving averaged AE_RIC_). The pink dots marks some distorted data (due to filtering edge effects).

**Figure 24 sensors-23-00488-f024:**
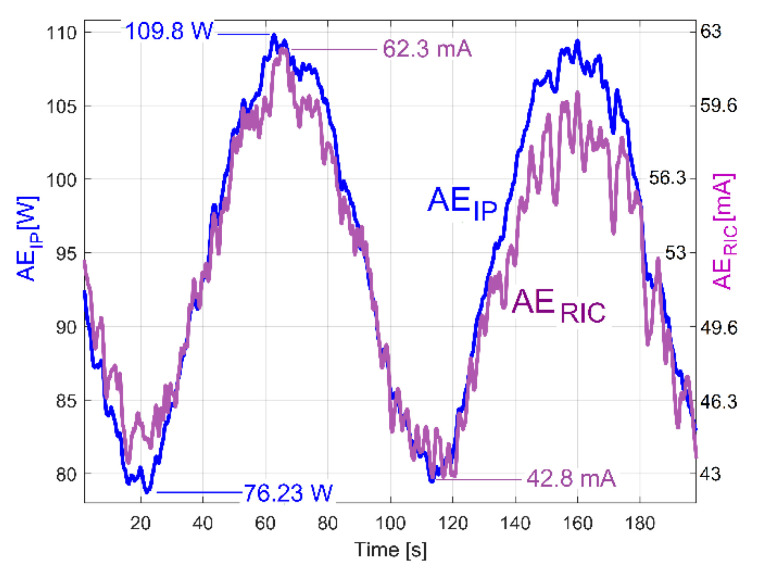
The evolutions of averaged AE_IP_ and AE_RIC_ related to the beating phenomenon.

**Figure 25 sensors-23-00488-f025:**
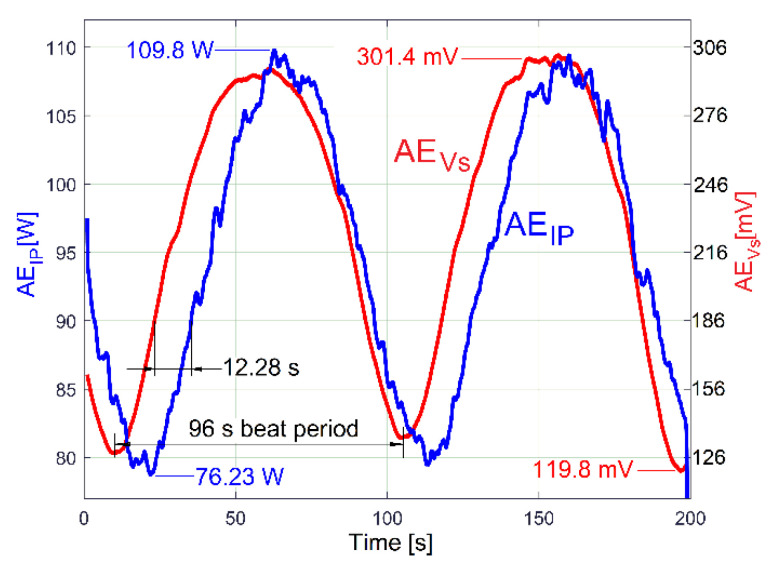
The evolutions of averaged AE_IP_ and AE_Vs_ related with the beating phenomenon.

**Figure 26 sensors-23-00488-f026:**
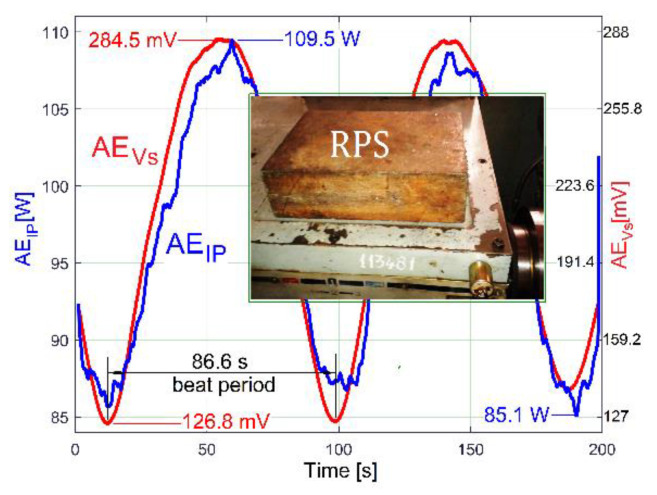
The averaged AE_IP_ and AE_Vs_ evolutions with an additional mass (RPS, 36.8 Kg) placed on gearbox.

**Figure 27 sensors-23-00488-f027:**
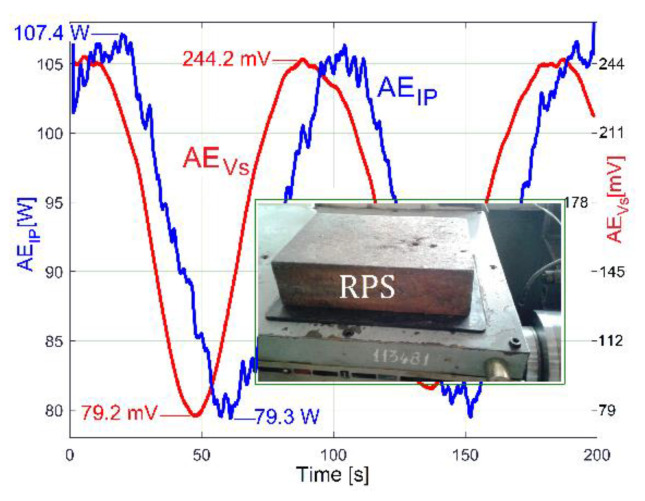
The averaged AE_IP_ and AE_Vs_ evolutions with a rubber plate placed between the additional mass RPS and the gearbox.

**Figure 28 sensors-23-00488-f028:**
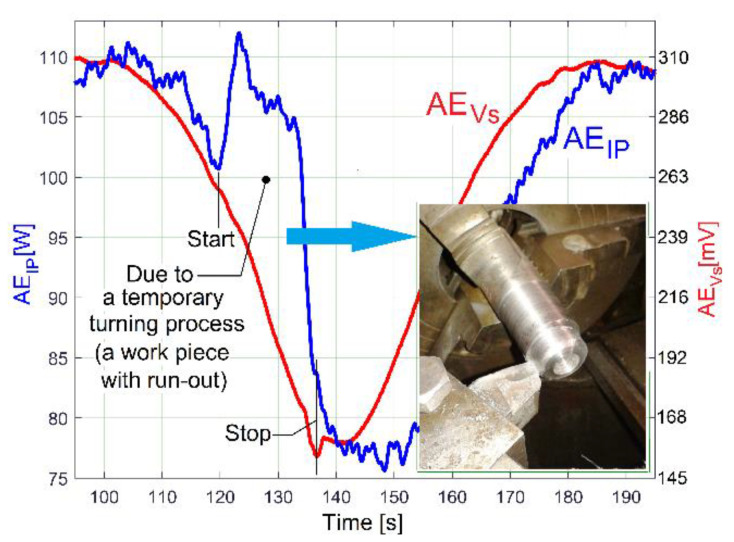
The influence of an interrupted longitudinal turning process on averaged AE_IP_ evolution.

**Figure 29 sensors-23-00488-f029:**
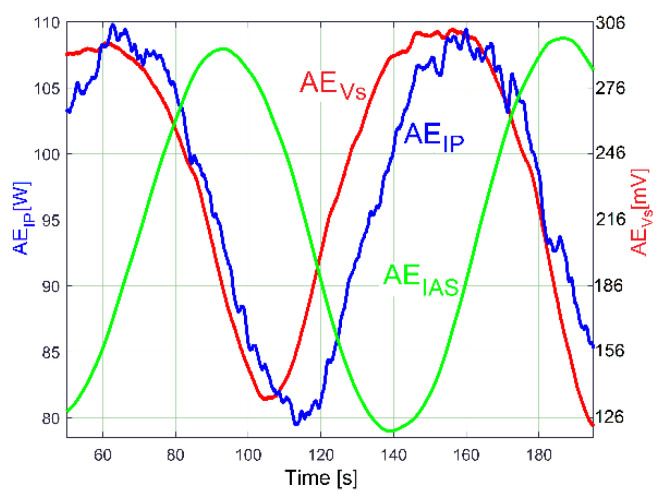
The evolutions of AE_IAS_, averaged AE_IP_ and AE_Vs_ (with large shift of phase between AE_IAS_ and AE_IP_).

**Figure 30 sensors-23-00488-f030:**
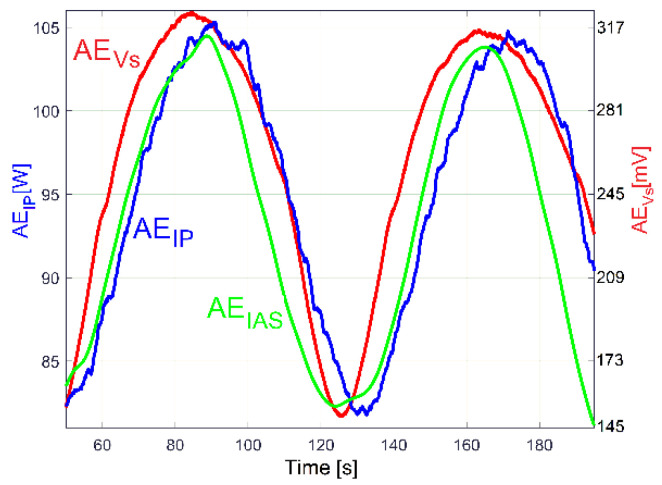
The evolutions of AE_IAS_, averaged AE_IP_ and AE_Vs_ (with small shift of phase between AE_IAS_ and AE_IP_).

**Figure 31 sensors-23-00488-f031:**
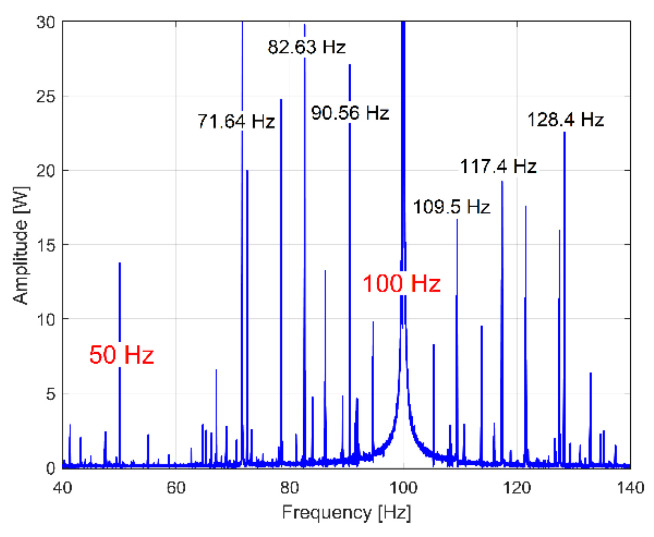
A short sequence of the IP spectrum (40 ÷ 140 Hz).

**Figure 32 sensors-23-00488-f032:**
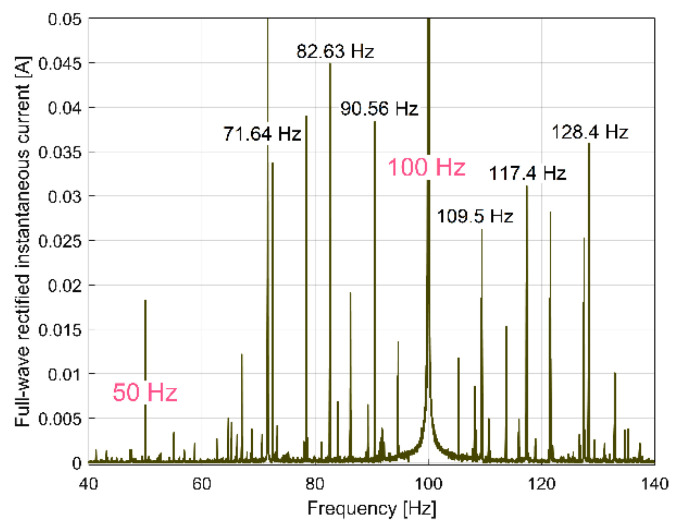
A short sequence of the RIC spectrum (40 ÷ 140 Hz).

**Figure 33 sensors-23-00488-f033:**
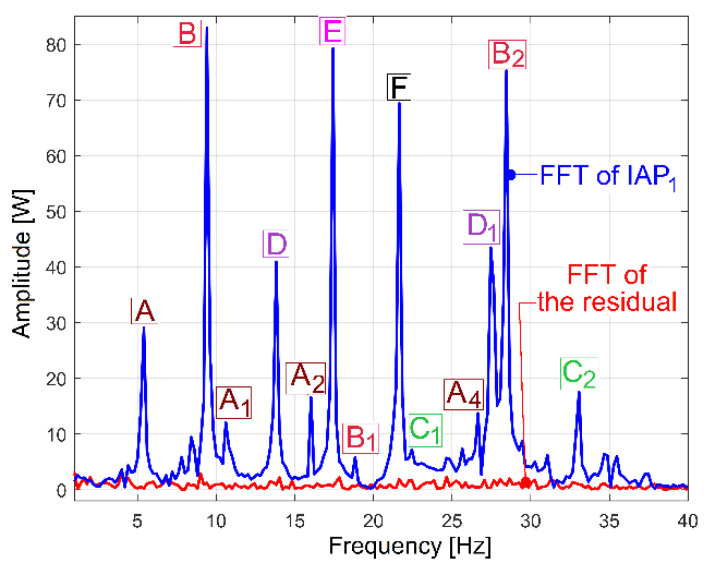
A comparison between the FFT of IAP_1_ and the FFT of the residual.

**Figure 34 sensors-23-00488-f034:**
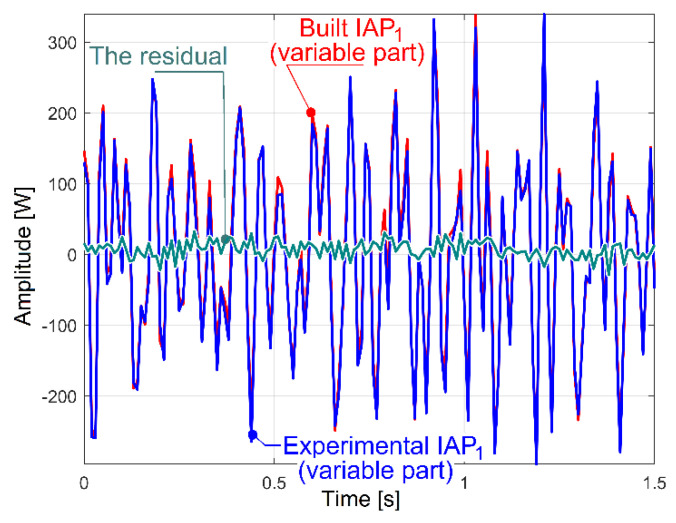
A short experimental and built IAP_1_ evolution.

**Figure 35 sensors-23-00488-f035:**
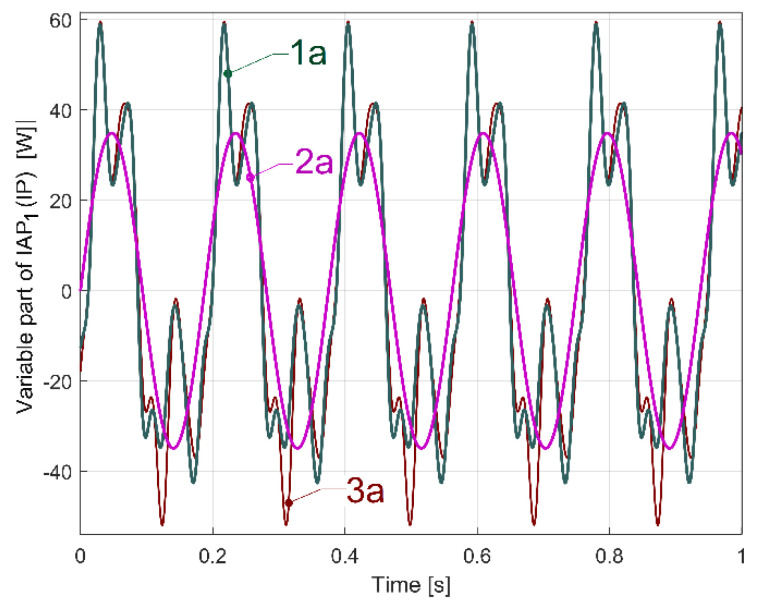
The evolution of variable part of IAP1 (IP) generated by flat belt 1.

**Figure 36 sensors-23-00488-f036:**
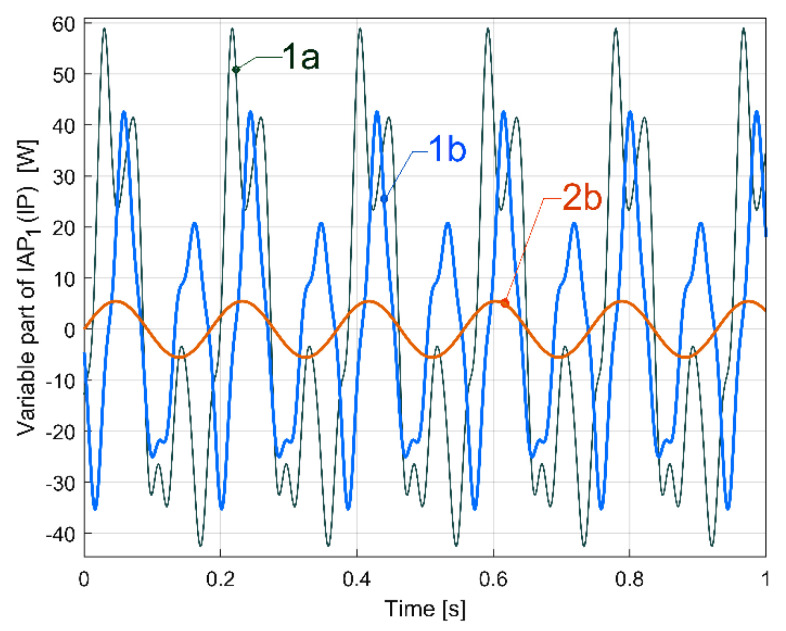
A comparison of the evolution of variable parts of IAP1 (IP) generated by flat belt 1 in two different circumstances.

**Figure 37 sensors-23-00488-f037:**
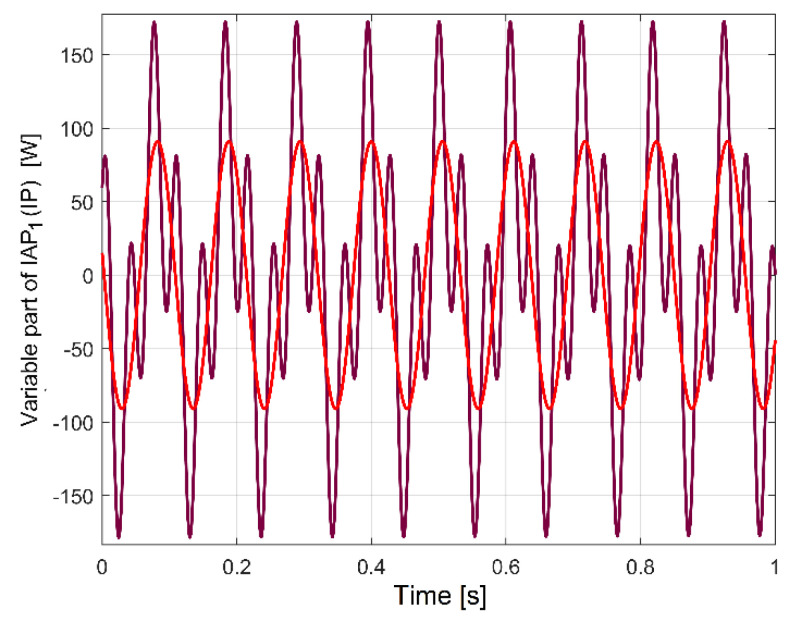
The evolution of variable part of IAP_1_ (IP) generated by flat belt 2 (coloured in a shade of magenta). The red coloured evolution describes the fundamental B.

**Figure 38 sensors-23-00488-f038:**
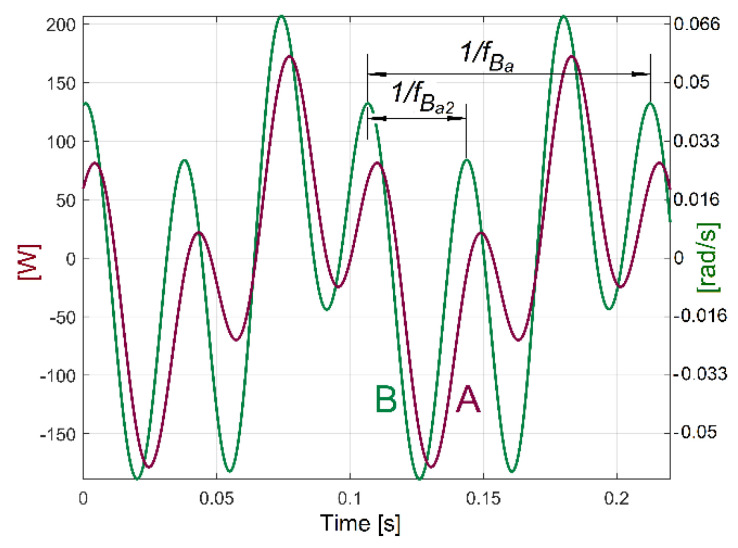
A comparative evolution of the variable part generated by flat belt 2 in IAP_1_ (curve A) and in IAS (curve B).

**Figure 39 sensors-23-00488-f039:**
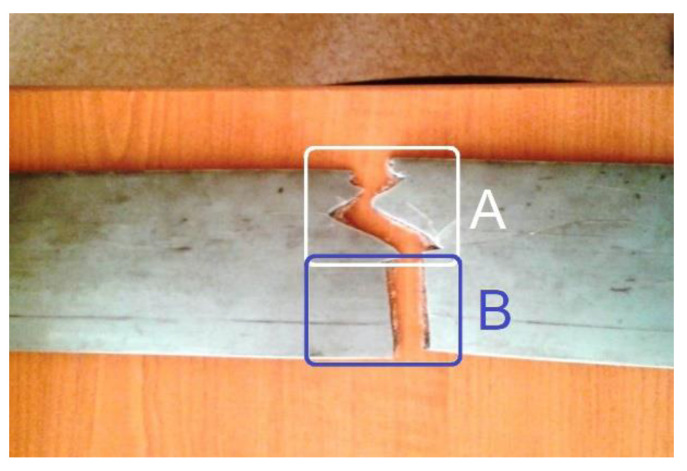
A view on a damaged flat belt 1.

**Figure 40 sensors-23-00488-f040:**
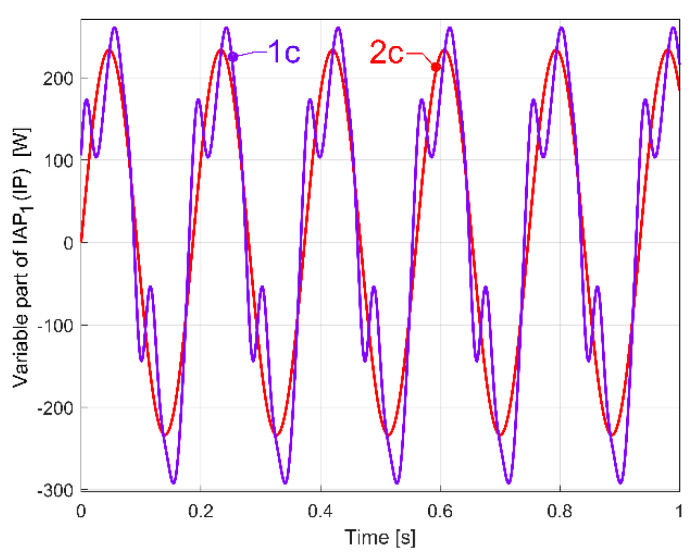
The evolution of variable part of IAP1 (IP) generated by a damaged flat belt 1 (with a severe tear in A, Figure 39).

**Figure 41 sensors-23-00488-f041:**
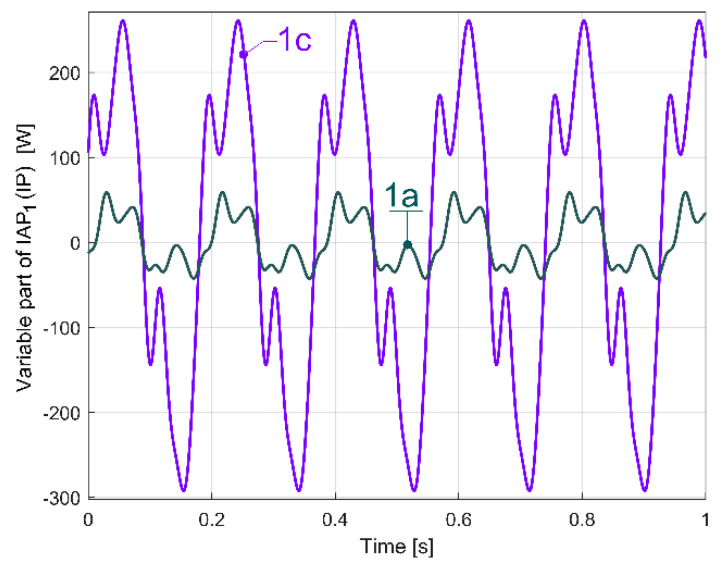
A comparison of variable parts of IAP_1_ (IP) generated by a damaged flat belt 1 (curve 1c) and a regular one (curve 1a).

**Figure 42 sensors-23-00488-f042:**
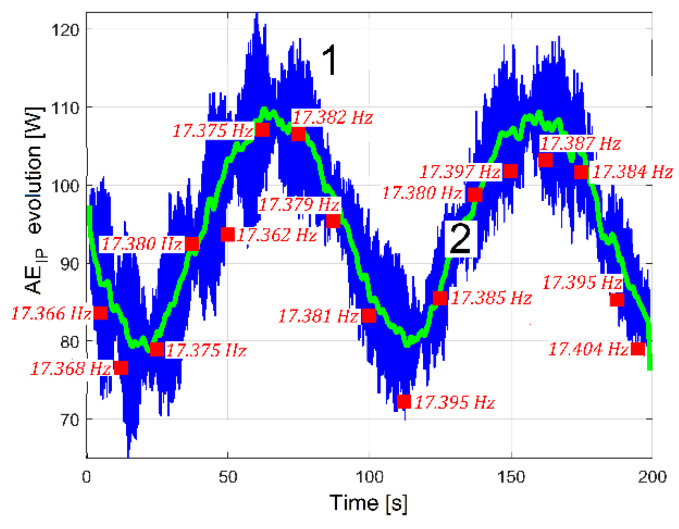
A completion of the Figure 22 with some equidistant timewise AE_PCF_ samples (as red rectangles).

**Figure 43 sensors-23-00488-f043:**
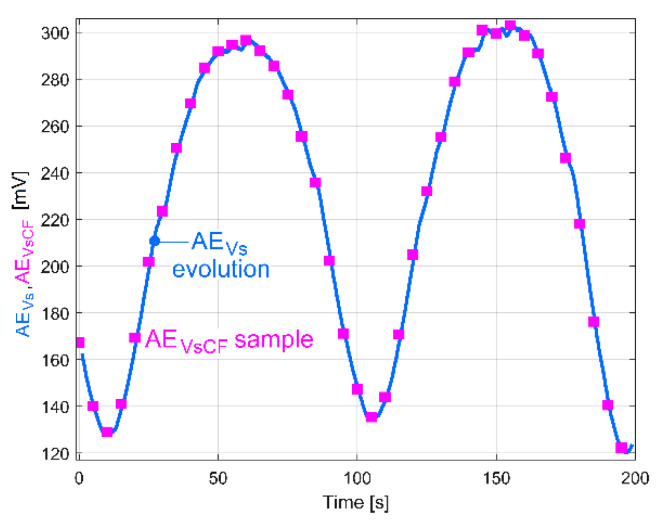
The evolution of AE_Vs_ and some equidistant timewise AE_VsCF_ samples (as pink rectangles).

**Figure 44 sensors-23-00488-f044:**
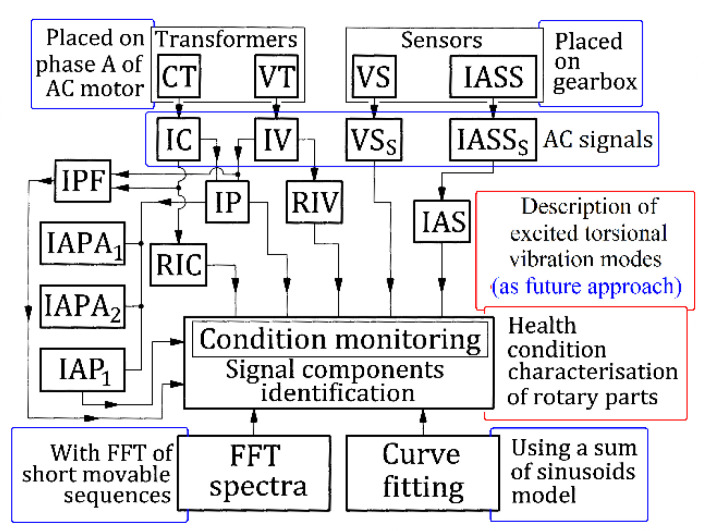
A block diagram related to signal processing for condition monitoring in this work.

**Figure 45 sensors-23-00488-f045:**
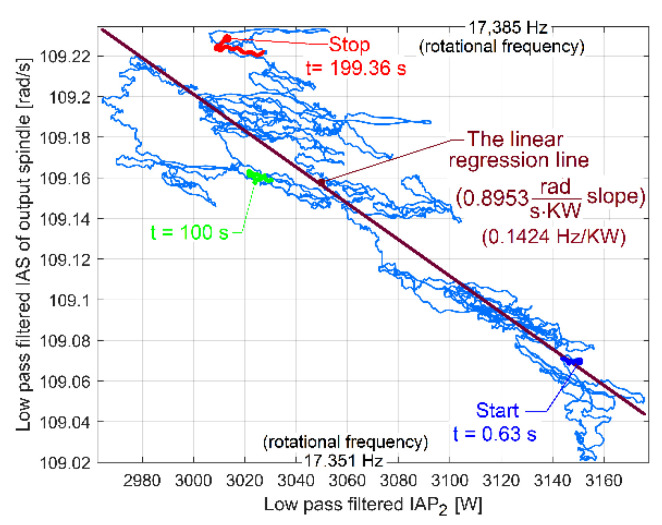
The low pass filtered IAS evolution (at the output spindle) depending on the low pass filtered AEP_2_ evolution.

**Figure 46 sensors-23-00488-f046:**
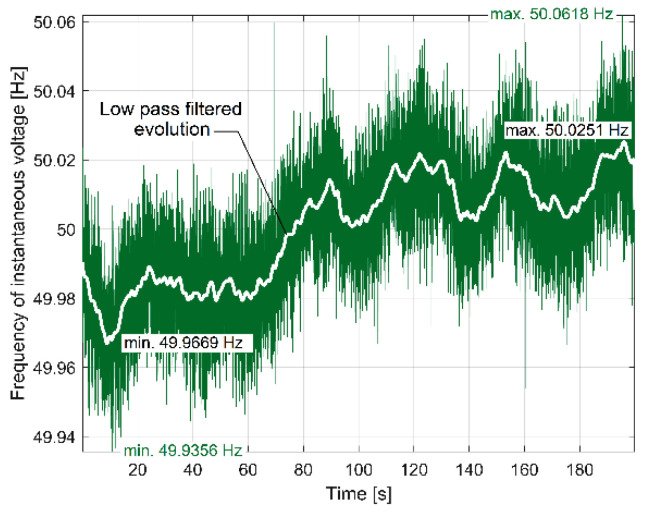
The evolution of frequency of instantaneous voltage on electrical supplying system depending by time.

**Table 1 sensors-23-00488-t001:** The description of the major components from FFT spectrum of IP (Figure 9).

IP Spectrum Component		Frequency [Hz]	Amplitude [W]	Describes the Behaviour of This Gearbox Part:
A (fundamental)		** *f_A_* ** **= 5.34**	**32.482**	**Flat belt 1** (5 years old, depicted in Figure 10) These components are involved in the description of flat belt 1 behaviour inside the first belt transmission.
Harmonics (H)	A_1_	*f_A_*_1_ = 10.68	10.750	
	A_2_	*f_A_*_2_ = 16.01	13.400	
	A_3_	*f_A_*_3_ = 21.35	6.390	
	A_4_	*f_A_*_4_ = 26.69	7.348	
	A_5_	*f_A_*_5_ = 32.02	1.483	
	A_6_	*f_A_*_6_ = 37.36	4.168	
B		** *f_B_* ** **= 9.45**	**85.016**	**Flat belt 2** (more than 40 years old, partially depicted in the upper part of Figure 10) These components are involved in the description of flat belt 2 behaviour inside the second belt transmission.
(H)	B_1_	*f_B_*_1_ = 18.90	4.138	
	B_2_	*f_B_*_2_ = 28.36	**77.010**	
	B_3_	*f_B_*_3_ = 37.80	0.637	
C		** *f_C_* ** **= 10.98**	**2.585**	**A v-belt used to drive a lubrication pump** (depicted in lower part of Figure 10) This pump (not highlighted in Figure 2) is driven by the same AC motor.
(H)	C_1_	*f_C_*_1_ = 21.97	3.937	
	C_2_	*f_C_*_2_ = 32.95	21.173	
D		** *f_D_* ** **= 13.75**	**34.378**	**The shaft II** 825 rpm rotational speed (60·*f_D_*)
(H)	D_1_	*f_D_*_1_ = 27.51	62.198	
E		** *f_E_* ** **= 17.37**	**73.126**	**Shaft III and the spindle** A study on this issue is presented below 1042.2 rpm rotational speed (60*·f_E_*)
(H)	E_1_	*f_E_*_1_ = 34.75	6.718	
F		** *f_F_* ** **= 21.56**	**74.749**	**The shaft I** 1293.6 rpm rotational speed (60·*f_F_*)
G		** *f_G_* ** **= 24.74**	**2.22**	**AC motor shaft** 1484.4 rpm rotational speed (60*·f_G_*)

**Table 2 sensors-23-00488-t002:** The description of some significant sinusoidal components from IP and IAP_1_.

Description of IP Sine Components from FFT Spectrum (from Table 1)				Description of IAP_1_ sine Components Identified by Curve Fitting		
Component Label		Frequency [Hz]	Amplitude [W]	Frequency [Hz]	Amplitude [W]	Phase at Origin of Time [rad]
A		5.34	32.482	5.336	34.88	−0.3943
Harmonics (H)	A_1_	10.68	10.750	10.664	15.48	−2.335
	A_2_	16.01	13.400	15.995	14.95	−1.437
	A_3_	21.35	6.390	21.310	5.504	2.709
	A_4_	26.69	7.348	26.674	9.172	0.3216
	A_5_	32.02	1.483	32.037	1.577	0.4805
	A_6_	37.36	4.168	37.330	4.183	4.981
B		9.45	85.016	9.4458	90.91	−3.311
(H)	B_1_	18.90	4.138	18.859	5.31	1.022
	B_2_	28.36	77.010	28.345	88.63	0.4546
	B_3_	37.80	0.637	37.783	0.833	1.5644
C		10.98	2.585	10.981	1.836	−4.022
(H)	C_1_	21.97	3.937	21.979	4.901	−1.341
	C_2_	32.95	21.173	32.929	20.82	2.14
D		13.75	34.378	13.747	46.19	−1.677
(H)	D_1_	27.51	62.198	27.501	63.02	−0.3428
E		17.37	73.126	17.363	84.08	1.524
(H)	E_1_	34.75	6.718	34.711	9.721	4.037
F		21.56	74.749	21.549	75.74	1.576
G		24.75	2.22	24.748	3.548	0.8004

## Data Availability

Not applicable.

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
