# Peer review of "On the Behaviour of an AC Induction Motor as Sensor for Condition Monitoring of Driven Rotary Machines"

_sensors, 2023, doi:10.3390/s23010488_

Round 1
Reviewer 1 Report
In this paper, authors presented the possible condition monitoring approach by employing AC induction motor as sensor. Overall, work is very well presented. The abstract can be improved by further briefing on what has been achieved in this work and its prospect applications. In introduction section, authors have repeatedly used the bracket () for explanation. This may be reduced by adjusting it within normal sentences. It is unclear how the instantaneous power may be distinguished from the noise? and its time length identification is a challenge or may be a difficult to detect or generalize. Therefore, it may be explained in the paper. The conclusion provided is very detailed. It may be split into two parts like discussion and conclusion or another appropriate way.
Reviewer 2 Report
This paper is very well written, the theory is supported by analytical reasoning and experimental results.
I would like to see more data on the parameters of the induction motor and types of gearbox failures that can be detected using this approach.
As we know, induction motor slip factor varies nonlinearly versus speed. So please clarify why not using synchronous machines instead of induction for this application.
Can you please provide some comments on the motor speed limitation if we use this method for gearbox condition monitoring.
